# Mitofusin 2 displays fusion-independent roles in proteostasis surveillance

Mariana Joaquim [1,2,3,11], Selver Altin[1,2,11], Maria-Bianca Bulimaga[1,2,3,4], Tânia Simões[1,2], Hendrik Nolte[1,2,10], Verian Bader [5], Camilla Aurora Franchino[2,3,6], Solenn Plouzennec[7], Karolina Szczepanowska[2,8], Elena Marchesan[9], Kay Hofmann [1], Marcus Krüger [1,2,3], Elena Ziviani[9], Aleksandra Trifunovic[2,3], Arnaud Chevrollier[7], Konstanze F. Winklhofer [5], Elisa Motori[2,3,6], Margarete Odenthal [3,4] & Mafalda Escobar-Henriques [1,2,3] ✉

Mitochondria are essential organelles and their functional state dictates cellular proteostasis. However, little is known about the molecular gatekeepers involved, especially in absence of external stress. Here we identify a role of MFN2 in quality control independent of its function in organellar shape remodeling. MFN2 ablation alters the cellular proteome, marked for example by decreased levels of the import machinery and accumulation of the kinase PINK1. Moreover, MFN2 interacts with the proteasome and cytosolic chaperones, thereby preventing aggregation of newly translated proteins. Similarly to MFN2-KO cells, patient fibroblasts with MFN2-disease variants recapitulate excessive protein aggregation defects. Restoring MFN2 levels re-establishes proteostasis in MFN2-KO cells and rescues fusion defects of MFN1-KO cells. In contrast, MFN1 loss or mitochondrial shape alterations do not alter protein aggregation, consistent with a fusion-independent role of MFN2 in cellular homeostasis. In sum, our findings open new possibilities for therapeutic strategies by modulation of MFN2 levels.

A key feature of mitochondria is the highly dynamic capacity to remodel their own shape, metabolic state and proteome composition[1,2]. This plasticity empowers mitochondria to quickly integrate and respond to both internal and external cues, allowing cellular stress management and ultimately determining cell fate[3–7]. Several recent studies provided enormous progress in the interplay between mitochondria and cytoplasmic machinery in cellular quality control[8–12]. For example, the importance of the mitochondrial import machinery for cellular homeostasis is nowadays commonly acknowledged[13–15]. On one hand, an active role of mitochondria in buffering cytotoxicity was proposed, for example by harnessing protein import of aggregation-prone cytosolic proteins, allowing their degradation by proteases present in the mitochondrial matrix[16–19]. On the other hand, perturbations in mitochondrial import are a challenge for cellular protein homeostasis, triggering cytosolic proteostasis networks[20–24]. For example, defective import of proteins into mitochondria can result in their cytosolic aggregation[25,26], which also promotes the aggregation of non-mitochondrial proteins in the

[1]Institute for Genetics, University of Cologne, Cologne, Germany. [2]Cologne Excellence Cluster on Cellular Stress Responses in Aging-Associated Diseases (CECAD), University of Cologne, Cologne, Germany. [3]Center for Molecular Medicine Cologne (CMMC), University of Cologne, Cologne, Germany. [4]Institute of Pathology, Medical Faculty of the University of Cologne and University Hospital of Cologne, Cologne, Germany. [5]Department Molecular Cell Biology, Institute of Biochemistry and Pathobiochemistry, Ruhr University Bochum, Germany, and Cluster of Excellence RESOLV, Bochum, Germany. [6]Institute for Biochemistry, University of Cologne, Cologne, Germany. [7]University of Angers, MitoLab Team, MitoVasc Unit, CNRS UMR6015, INSERM U1083, SFR ICAT, Angers, France. [8]ReMedy International Research Agenda Unit, International Institute of Molecular Mechanisms and Machines (IMol), Polish Academy of Sciences, 00-783, Warsaw, Poland. [9]Deparment of Biology, University of Padova, Padova, Italy. [10]Present address: MPI for Biology of Ageing, 50931 Cologne, Germany. [11]These authors contributed equally: Mariana Joaquim, Selver Altin. ✉e-mail: Mafalda.Escobar@uni-koeln.de

cytoplasm, ultimately creating proteotoxicity[27,28]. This leads to the subsequent sequestration of chaperones that are necessary for mitochondrial import, further enhancing mitochondrial dysfunction and cellular toxicity[29]. Unsurprisingly, protein import defects can induce mitophagy, a selective pathway allowing elimination of the damaged organelles[30].

Morphological adaptations of mitochondria are also intimately linked to the cellular state[2]. Mitochondrial remodeling is enabled by fusion and fission events, performed by Dynamin-Related GTPase Proteins (DRPs)[31,32]. The DRPs responsible for fusion of the mitochondrial outer membrane are called mitofusins, represented by the homologous proteins MFN1 and MFN2[32]. Mitofusins, while being integral membrane proteins, are almost entirely exposed to the cytoplasm, a perfect location for a relay point in information exchange between mitochondria and their cellular environment[9,33,34]. Disruption of mitochondrial fusion by loss of mitofusins leads to mitochondrial fragmentation and causes severe impairment of oxidative phosphorylation and mitochondrial respiration[35–43]. Low levels of MFN2 and respiratory defects were also observed in the context of obesity[44,45] and are associated with diabetes[46,47], non-alcoholic fatty liver disease[48,49], cancer[50], cardiac dysfunction[51], and neurodegeneration[52–55]. Notably, point mutations in the *Mfn2* gene are causative of the neurodegenerative disease Charcot Marie-Tooth subtype 2 A (CMT2A)[56–58]. CMT is an incurable peripheral neuropathy characterized by progressive distal muscle weakness, sensory abnormalities and muscular atrophy[59]. *Mfn2* mutations account for 90% of the most severe cases of CMT2A[60], with approximately 170 amino acid mutations in MFN2 identified so far. In contrast, there are no *Mfn1* mutations known to cause disease.

While the impact of MFN2 in mitochondrial fusion and cellular respiration is well described[37], the mechanisms at the basis of its disease gate-keeping roles are poorly understood. Here, using human cell line models, we identify a broad effect of MFN2 in the cellular proteome composition and quality control, beyond its canonical role in mitochondrial fusion. Cells lacking MFN2 present decreased levels of import components, accumulate unimported Ser/Thr kinase PTEN-induced putative kinase 1 (PINK1) and present a mild increase in mitophagy. Moreover, MFN2 interacts with quality control factors, including cytosolic chaperones and the proteasome. Consistently, absence of MFN2 causes protein aggregation, a feature also observed in CMT2A primary human fibroblasts. Finally, inhibition of protein translation in MFN2 KO cells abrogates protein aggregation. Thus, we propose that MFN2 acts as a cellular sensor at the mitochondrial surface, thereby controlling homeostasis of newly synthesized proteins, highlighting a disease-relevant function of MFN2.

## Results

### MFN2 rescues mitochondrial fusion defects of MFN1 and MFN2 KO cells

MFN1 and MFN2 have redundant but not totally overlapping functions, which are not entirely characterized. To study the role of each mitofusin, we generated single knockouts (KOs) in human cell culture models, including HEK293 (Supplementary Fig. 1a). Mitochondrial morphology was analyzed via immunostaining with antibodies against the outer mitochondrial membrane protein TOM20 and the inner mitochondrial membrane protein ATP5β, which confirmed that absence of mitofusins causes mitochondrial fragmentation (Fig.1a). However, ablation of MFN1 or MFN2 disrupted the mitochondrial network in different ways. Whereas MFN1 KO cells (1KO) presented small dot-shaped and dispersed mitochondrial fragments, but with no major ultrastructural changes, in MFN2 KO cells (2KO) mitochondrial fragments were much larger, clustered and swollen (Fig. 1a, Supplementary Fig. 1b, c). As previously observed[61,62], the levels of mitochondrial DRPs are interdependent (Supplementary Fig. 1d, e). MFN1 and MFN2 KO were also created in HeLa cell lines (Supplementary Fig. 2a), which confirmed the differential contribution of MFN1 and

MFN2 for mitochondrial morphology and ultrastructure (Supplementary Fig. 2b–d). The mild differences observed between the two cellular models, for example better preservation of cristae morphology in HEK293 cells, likely reflects cell-type specific differences. Taken together, our results confirm the need of both mitofusins for mitochondrial fusion.

To better compare both mitofusins, we created MFN1 and MFN2 double KO cells (DKO), which showed perinuclear clustered mitochondria, resembling 2KO cells (Supplementary Figs. 1, 2). Consistent with a stronger impact of MFN2-loss over MFN1-loss, 2KO cells showed an approximately 30% reduction of cellular proliferation (Supplementary Fig. 3a). This further supports different roles for each mitofusin. Importantly, re-expression of MFN1 in 1KO cells and of MFN2 in 2KO cells restored mitochondrial morphology (Supplementary Fig. 3b), which also demonstrates the functionality of the plasmid-encoded FLAG-tagged proteins. Strikingly however, MFN2 also rescued mitochondrial fragmentation in 1KO and DKO cells, while MFN1 expression did not alter the morphology defect present in 2KO and DKO. In conclusion, MFN2 alone is sufficient to rescue the mitochondrial morphology defects caused by loss of both mitofusins.

### MFN2 ablation profoundly alters the cellular proteome

To further characterize the impact of MFN2 loss, we performed a label-free quantification by mass spectrometry of whole protein extracts and compared 2KO cells to WT and 1KO. Moreover, we analyzed 2KO cells stably expressing MFN2 to endogenous-like levels (2 + 2 cells, Supplementary Fig. 4a). We measured more than 6.1 K protein groups, including more than 788 proteins of the Mito-Carta3.0 (Supplementary Data 1). To obtain a systematic overview of the high dimensional data, we performed a principal component analysis (PCA) and observed that 2KO cells segregated from WT, 1KO and 2 + 2 cells (Fig. 1b). Ablation of MFN2 significantly altered 1421 proteins, most of those (912; 64%) were downregulated and vastly located to mitochondria(482; 53%) (Supplementary Fig. 4b). Consistently, MFN2 ablation led to an overall loss of the mitochondrial mass (Fig. 1c and Supplementary Fig. 4c, left panel). In contrast, among the 509 upregulated proteins (36%) only 29 (6%) are mitochondrial. Therefore, while most of the proteome changes reflect a decrease of mitochondrial proteins, the over 30% of the proteome that was upregulated largely affected non-mitochondrial (94%) proteins (Supplementary Fig. 4d). Importantly, re-expressing MFN2 rescued these alterations, allowing to exclude clonal effects (Fig. 1c, Supplementary Fig. 4b, c). To systematically identify mitochondrial pathways regulated in 2KO cells independently of the decreased mitochondrial mass, we normalized the mitochondrial mass in all samples (Supplementary Fig. 4c, right panel) and performed a 1D enrichment analysis. MFN2 KO cells presented a significant downregulation of the mitochondrial ribosome and OXPHOS subunits and upregulation of the fatty acid oxidation and ROS & glutathione pathways (Fig. 1d, e, Supplementary Fig. 4e, f). In conclusion, absence of MFN2 causes profound proteome rewiring events.

Hierarchical clustering analysis of the proteomics data, as well as Western blot, native gel electrophoresis and oxygen consumption experiments, confirmed the strong impairment in OXPHOS protein levels, complex assembly and respiratory capacity in 2KO cells (Supplementary Fig. 5), consistent with previous observations[32,37]. These defects were rescued by stable expression of MFN2 (Supplementary Fig. 6a–c). Nevertheless, no major alterations in the mitochondrial membrane potential could be observed (Supplementary Fig. 6d). Finally, mRNA levels of the OXPHOS subunits were not changed in 2KO cells and their protein levels were unaffected by proteasomal inhibition (Supplementary Fig. 6e, f). In summary, MFN2 has a significant influence on proteome composition, mitochondrial respiration and proliferative capacity. This expands the differences

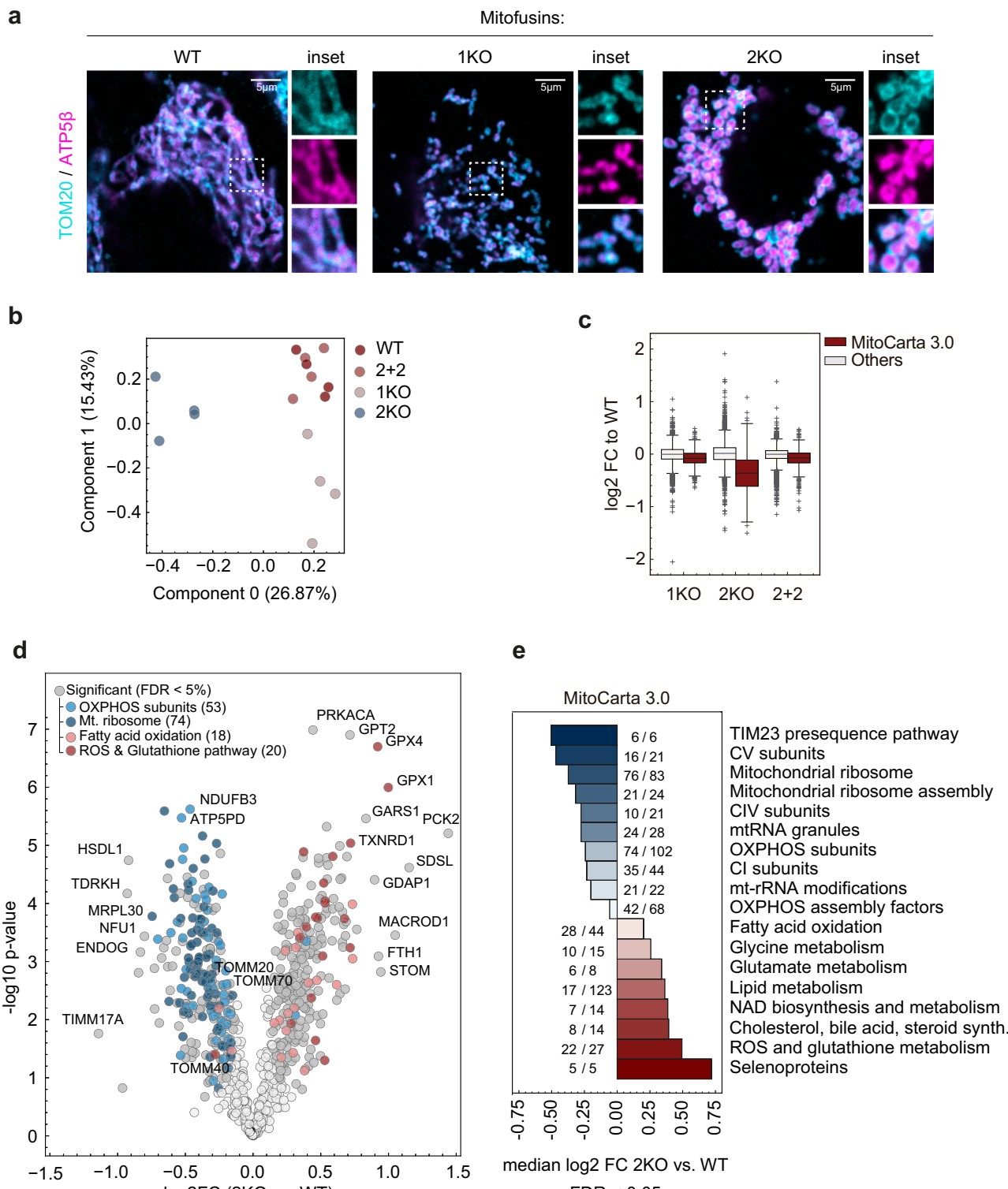

**a** Mitofusins:
WT / inset / 1KO / inset / 2KO / inset
TOM20 / ATP5β

**b** Component 1 (15.43%) vs Component 0 (26.87%); WT, 2+2, 1KO, 2KO

**c** log2 FC to WT; 1KO, 2KO, 2+2; MitoCarta 3.0, Others

**d** -log10 p-value vs log2FC (2KO vs. WT)
Significant (FDR < 5%)
OXPHOS subunits (53)
Mt. ribosome (74)
Fatty acid oxidation (18)
ROS & Glutathione pathway (20)

**e** MitoCarta 3.0
median log2 FC 2KO vs. WT
FDR < 0.05

| Count | Pathway |
|---|---|
| 6 / 6 | TIM23 presequence pathway |
| 16 / 21 | CV subunits |
| 76 / 83 | Mitochondrial ribosome |
| 21 / 24 | Mitochondrial ribosome assembly |
| 10 / 21 | CIV subunits |
| 24 / 28 | mtRNA granules |
| 74 / 102 | OXPHOS subunits |
| 35 / 44 | CI subunits |
| 21 / 22 | mt-rRNA modifications |
| 42 / 68 | OXPHOS assembly factors |
| 28 / 44 | Fatty acid oxidation |
| 10 / 15 | Glycine metabolism |
| 6 / 8 | Glutamate metabolism |
| 17 / 123 | Lipid metabolism |
| 7 / 14 | NAD biosynthesis and metabolism |
| 8 / 14 | Cholesterol, bile acid, steroid synth. |
| 22 / 27 | ROS and glutathione metabolism |
| 5 / 5 | Selenoproteins |

between MFN1 and MFN2 loss and suggests a role of MFN2 beyond mitochondrial fusion.

## MFN2 controls the import machinery

Besides alterations in mitochondrial translation and respiration, MFN2 depletion also reduced the levels of the import machinery (Fig. 1d, e). This includes the outer membrane translocase components TOM20, TOM40 and TOM70 (Fig. 2a and Supplementary Fig. 7a). Similar results were observed in mitochondrial cell extracts (Supplementary Fig. 7b), confirming an effect beyond the general mitochondrial mass decrease

caused by MFN2 loss and consistent with the observations after mitochondrial mass normalization (Fig. 1d). To reinforce these results, we cultivated WT and 2KO embryonic murine neurons. Notably, this showed reduced TOM20 levels upon ablation of MFN2 in the neuronal cell culture system (Fig. 2b), highlighting a conserved effect of MFN2 in the import machinery in different organisms and cell types. These observations raised the possibility of deficient protein import into mitochondria in the absence of MFN2. To test this hypothesis, we analyzed the levels of the mitochondrial Ser/Thr kinase PINK1, an acknowledged marker of import defects[63]. PINK1 is normally imported

**Fig. 1 | MFN2 controls mitochondrial mass and composition. a** Confocal images after immunostaining with the inner and outer mitochondrial membrane proteins ATP5β (in magenta) and TOM20 (in cyan), respectively, of HEK WT or corresponding knockout cells for MFN1 (1KO) or MFN2 (2KO). Scale bar: 5 μm. Insets of white dotted boxes are shown on the right side of each image. **b** Principal Component Analysis (PCA) of proteomes of HEK WT, 1KO, 2KO and 2KO cells stably expressing MFN2FLAG (2 + 2) (n = 4 biological replicates). Groupings are color-coded and the explained variance is indicated in brackets. **c** Boxplot analysis showing the log2 fold change distribution of 1KO, 2KO and 2 + 2 versus WT cells (n = 4 biological replicates) for MitoCarta 3.0 (in red) and non-MitoCarta 3.0 (in gray) protein groups. Boxes borders indicate the 25% and 75% quantiles and outliers are indicated

by a + sign (greater distance than 1.5 * inter quantile range). The minimum and maximum values are shown by the whiskers excluding outliers. **d** Volcano plot of the log2 fold change between 2KO versus WT cells (n = 4 biological replicates). The significantly different (Two-sided unpaired t-test followed by a permutation-based FDR correction FDR < 0.05, s0 = 0.1, #permutations = 500) protein groups of different MitoCarta 3.0 pathways are highlighted by color. **e** Bar graph displaying the result of the 1D Enrichment from the MitoCarta 3.0 based normalization data showing the median log2 fold change (all protein groups annotated by the given pathway) of 2KO versus WT cells (n = 4 biological replicates). The number of proteins identified within the total number of proteins in each pathway are annotated next to respective bar. FDR < 0.02. Source data are provided as Source Data file.

and cleaved by the inner membrane rhomboid protease PARL, the short form being subsequently exported to the cytoplasm and degraded by the proteasome[63]. However, upon mitochondrial damage or under conditions of deficient protein import, the full-length/unprocessed form of PINK1 accumulates at the outer mitochondrial membrane. Cleaved PINK1, which requires previous import of the full-length protein to mitochondria, can be detected upon proteasomal inhibition. Supporting defective mitochondrial import in 2KO cells, MFN2 loss led to an accumulation of full length PINK1 (Fig. 2c and Supplementary Fig. 7c) and a decrease in the cleaved form of PINK1 (Fig. 2d), without affecting PARL levels (Supplementary Fig. 7d). In turn, MFN2 loss did not affect mRNA levels of PINK1 or TOM20 (Supplementary Fig. 7e) and proteasomal inhibition did not alter the levels of TOM20 (Supplementary Fig. 7f). Again, these post-transcriptional effects were rescued to WT-like levels upon stable expression of MFN2.

Next, considering the reported role of MFN2 in mitochondria-endoplasmic reticulum (ER) tethering, we exogenously expressed a broadly used mitochondria-ER tether construct[64,65] but observed no alteration in the levels of PINK1, TOM20 or OXPHOS subunits (Fig. 2e and Supplementary Fig. 7g, h). Finally, we analyzed the physiological consequence of PINK1 increase in 2KO cells. PINK1 accumulation at the mitochondrial surface signals the removal of damaged organelles by mitophagy[30,55,63]. Consistently, qualitative analysis revealed the presence of autophagic bodies in 2KO cells, along with increased LC3 lipidation (LC3-II), observed in both total cell extracts and mitochondrial preparations (Supplementary Fig. 8a–c). To quantitatively assess mitophagy, we used the MtKeima reporter system, and targeted the pH sensitive fluorescent Keima protein to the mitochondrial matrix. By accounting for mitochondria engulfed by the lysosomes, this assay allows to assess completion of mitophagy. MtKeima measurements confirmed increased mitophagy in 2KO cells, both in basal conditions and upon inhibition of the deubiquitylase (DUB) USP14 with IU1, a treatment shown to induce mitophagy[66] (Fig. 2f). Together, these results point to a role of MFN2 in the regulation of mitochondrial protein import and PINK1 levels, with a mild impact on mitophagy, consistent with defective mitochondrial homeostasis in 2KO cells.

## MFN2 is ubiquitylated and binds the proteasome

To further investigate the differences between MFN1 and MFN2, we compared the protein interactome of both mitofusins, using stable cell lines transgenically expressing tetracycline inducible FLAG-tagged MFN1 in 1KO cells (1 + 1) and MFN2 in 2KO cells (2 + 2). While 2 + 2 cells expressed MFN2 to endogenous-like levels without addition of tetracycline (Supplementary Fig. 4a), 10 μg/μl tetracycline was added to 1 + 1 cells to induce MFN1 expression (Supplementary Fig. 9a). MFN1FLAG and MFN2FLAG were precipitated with anti-FLAG magnetic beads, eluted with 3xFLAG peptide and subjected to mass-spectrometry analysis (Fig. 3a, Supplementary Fig. 9b, c and Supplementary Data 2). Notably, the two mitofusins presented different interactome signatures. MFN2 bound to the ubiquitin-chaperone p97 and to 18 out of the 33 human proteasomal subunits, belonging to both the 20S (8 subunits, out of 14) and the 19S (10 subunits, out of 19) complexes (Fig. 3a and Supplementary Fig. 9b). In contrast, these interactions were not observed

for MFN1 (Supplementary Fig. 9c), despite its highly homologous sequence to MFN2. An orthogonal approach using proximity ligation assays (PLA) confirmed the interaction of endogenous MFN2 with the 20S and the 19S proteasome. No interaction with the proteasome was observed for MFN2 in 2KO cells, i.e. in presence of MFN1 (Fig. 3b, Supplementary Fig. 10a), and for MFN1, both in WT and in 2KO cells (Supplementary Fig. 10b), which ascertains the specificity of MFN2 and not MFN1 in proteasome binding. Despite these interactions, the protein levels of proteasomal subunits were not affected by the absence of MFN2 (Supplementary Fig. 10b).

To interrogate if MFN2 could affect proteasomal assembly and thereby its activity, we tested reported proteasomal-dependent responses to mitochondrial defects that are also observable in 2KO cells. We analyzed transcriptional alterations in the proteasomal subunits PSMD9, PSMD10, and PSMD11, previously shown to be downregulated upon defects in OXPHOS CI[67]. However, RT-qPCR revealed that 2KO cells did not mimic this effect (Supplementary Fig. 10c). On the other hand, defects in mitochondrial protein import were shown to increase proteasomal activity, via upregulation of the chaperone HSPB1, the translation elongation factor EEF1A2 and the immunoproteasome-specific subunit PSMB9[26]. However, 2KO cells also did not present altered levels of their mRNAs (Supplementary Fig. 10d). We then tested if protein import could affect the interaction between MFN2 and the proteasome. As monitored by PINK1 accumulation, dissipation of membrane potential with CCCP efficiently blocks protein import (Supplementary Fig. 10e). However, CCCP reduced the interaction between MFN2 and the proteasome (Supplementary Fig. 10f), consistent with its effect on MFN2 turnover[68,69]. Finally, external and acute proteostasis defects at the mitochondrial matrix were shown to lead to an upregulation of mtUPR, a stress-response pathway allowing mitochondrial-cytosolic communication[17]. However, 2KO cells did not upregulate the mtUPR markers *HSPD1* and *HSPA9*, encoding the mitochondrial chaperones HSP60 and HSP70, respectively (Supplementary Fig. 10d, Fig. 2c). Thus, MFN2 elicits a quality control response different from previously reported pathways.

Mitofusins were shown to be promptly ubiquitylated by several E3 ligases in response to stress and metabolic shifts[33]. The fact that MFN2 interacted with proteasomal subunits in absence of stress suggested that MFN2 could also be constitutively ubiquitylated. To test this hypothesis, an MFN2-specific antibody was used to immunoprecipitate endogenous MFN2. Immunoblotting with the same MFN2-specific antibody detected the unmodified MFN2 (arrowhead) as well as higher molecular weight bands (arrows). The same bands were recognized by the ubiquitin antibody and could be eliminated by treatment with USP21, a broad and very active DUB (Fig. 3c, d). Subsequent experiments in 2 + 2 cells transiently expressing MYC-tagged ubiquitin and treated with USP21 confirmed ubiquitylation of MFN2 under basal conditions in 2 + 2 cells (Supplementary Fig. 10g, h). This provided a rational basis for MFN2 interaction with the proteasome. Therefore, WT cells were treated with the ubiquitin activating (E1) UBA1 inhibitor MLN-7243, which depleted endogenous MFN2 ubiquitylation (Fig. 3e). Importantly, MLN-7243 treatment also significantly impaired MFN2 interaction with the proteasome

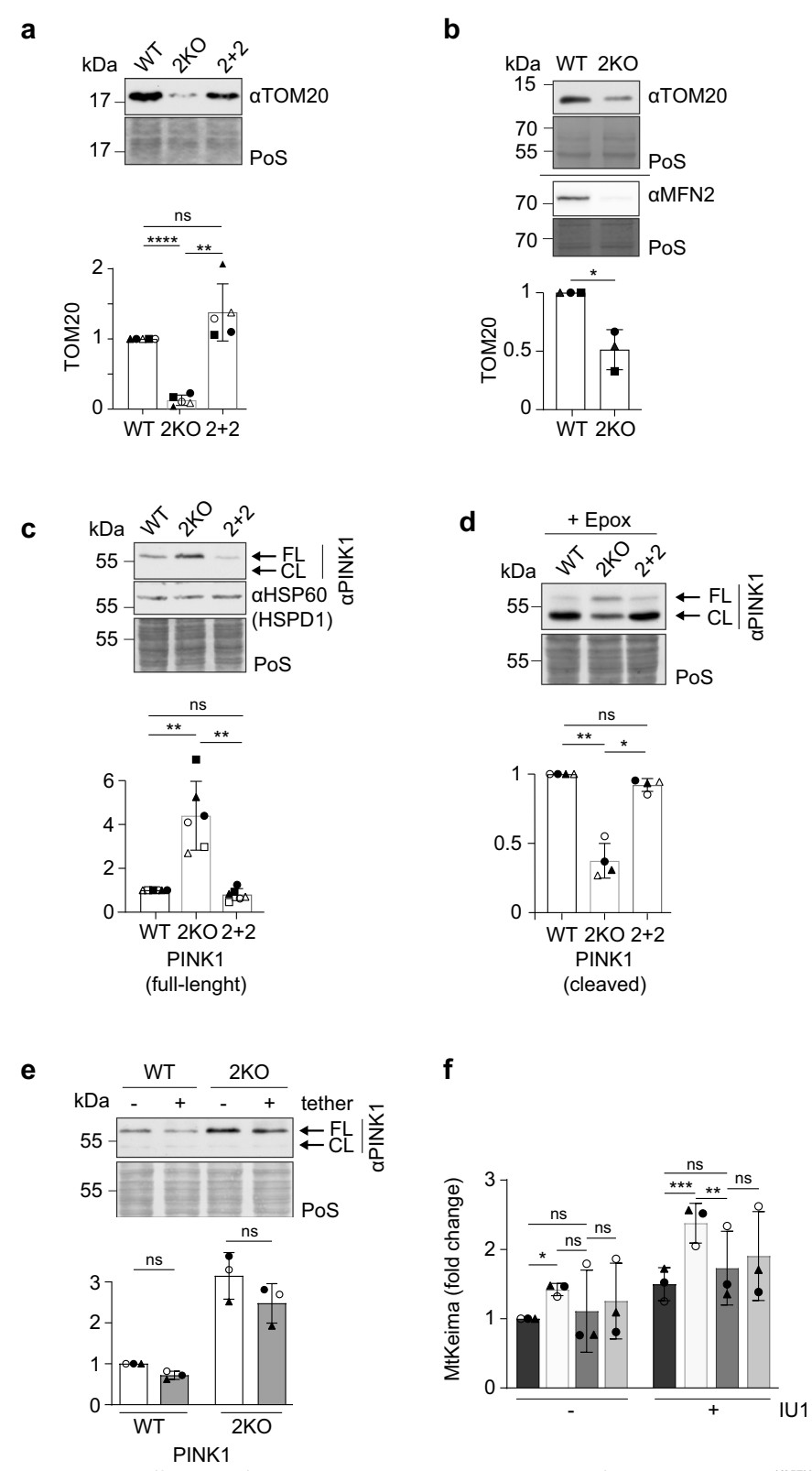

(Fig. 3f). In sum, ubiquitylated MFN2 binds the proteasome under constitutive growth conditions.

## MFN2 controls MFN1 turnover

MFN2 ubiquitylation and interaction with the proteasome suggested that MFN2 could regulate substrate turnover. Interestingly, both Western blot and immunostaining experiments revealed a

slight increase in protein ubiquitylation in 2KO cells (Supplementary Fig. 11a, b), suggesting a general but mild role of MFN2 in protein degradation. In addition, the enrichment of the ubiquitin signal was not restricted to mitochondria, suggesting broader effects (Supplementary Fig. 11b). Next, we analyzed the stability properties of mitofusins under constitutive growth conditions, after translation inhibition with CHX. We noticed that MFN2 is a mildly unstable

**Fig. 2 | MFN2 loss impairs protein import, increasing PINK1 levels. a** Western blot analysis of total cell lysates from HEK WT, 2KO and 2 + 2 cells, immunoblotted with anti-TOM20. Staining of total protein with PoS was used as loading control. Bars represent the mean fold change relative to WT ± SD (*n* = 5 biological repli-cates). Individual values of each experiment are represented in white or black filled squares triangles or circles. RM one-way ANOVA was applied. *P*-value left to right (p L-R): **** <0.001, ** 0.0087. **b** Western blot analysis as in (**a**) from primary cultures of WT or Mfn2 depleted (2KO) mouse cortical neurons, immunoblotted with anti-TOM20 and anti-MFN2. (*n* = 3 biological replicates). Paired t-test was applied. *P* value: 0.0388 **c** Western blot as in (**a**) of total cell lysates from HEK WT, 2KO and 2 + 2 cells, immunoblotted with anti-PINK1 (FL: full-length; CL: cleaved) and anti-HSP60. (*n* = 6 biological replicates). RM one-way ANOVA was applied. P L-R: ** 0.0074, ** 0.0041. **d** Western blot analysis as in a) of total cell lysates from HEK WT

and 2KO cells treated with epoxomicin (1 μM, 2 h), immunoblotted with anti-PINK1 (revealing its cleaved (CL) form). (*n* = 4 biological replicates). RM one-way ANOVA was applied. P L-R: ** 0.0044, * 0.015. **e** Western blot analysis as in (**a**) of total cell lysates from HEK WT and 2KO cells transiently transfected with a control vector (-) or with the artificial mito-ER tether (+), immunoblotted with anti-PINK1 (FL: full-length; CL: cleaved). (*n* = 3 biological replicates). Two-way ANOVA was applied. **f** MtKeima mitophagy measurement in HEK WT, 2KO, 2 + 2 and 2 + 2$^{K357N}$ cells treated with DMSO (-) or with IU1 (+, 100 μM). Bars represent mean fold change relative to DMSO-treated WT (*n* = 3 biological replicates) ± SD. Individual values of each experiment are discriminated in black filled triangles and white or black filled circles. Two-way ANOVA was applied. P L-R: * 0.0227, *** 0.0005, ** 0.0024. Source data are provided as Source Data file.

protein, whose turnover can be blocked by proteasomal inhibition (Supplementary Fig. 11c, d), confirming previous observations[70]. The same was observed for MFN1 (Supplementary Fig. 11c, e). Surprisingly, absence of MFN2 prevents basal turnover of MFN1 (Fig. 4a, Supplementary Fig. 11c. Nevertheless, we did not observe an increase in the steady state levels of MFN1 in 2KO cells (Supplementary Fig. 11f). This is possibly the net result of both the role of MFN2 in promoting MFN1 turnover and the general decrease in mitochondrial mass observed in MFN2 KO cells (Fig. 1c).

To further explore the interplay between both mitofusins, we analyzed the role of MFN2 in MFN1 turnover under stress. We induced apoptotic cell death by treating cells with actinomycin D (ActD), which leads to DNA damage and thereby initiates the intrinsic apoptotic pathway, via BAX/BAK oligomerization. Time-dependent cleavage of the poly [ADP-ribose] polymerase 1 (PARP1) confirmed apoptosis induction (Fig. 4b). Notably, while ActD decreased the protein levels of both mitofusins in WT cells, absence of MFN2 clearly stabilized MFN1 (Fig. 4b). In contrast, MFN1 did not affect MFN2 turnover. Similarly to MFN1, apoptotic induction triggered degradation of PINK1 in a MFN2-dependent manner (Fig. 4c). In contrast, the induced-myeloid leuke-mia cell differentiation (MCL1) protein, known to be degraded upon apoptosis[71], was not stabilized in the absence of MFN (Fig. 4d). Moreover, under mitophagy conditions, 2KO cells did not prevent MFN1 turnover (Supplementary Fig. 12a). In addition, the E4 ubiquitin ligase UBE4B, recently shown to regulate MFN2 under stress[68], was not affected by MFN2 ablation (Supplementary Fig. 12b).

Next, we analyzed the effect of MFN2 on another mitochondrial protein - MARCH5, an E3 ligase located at the outer membrane. MARCH5 physically interacts with MFN2[72], as we could confirm by the immunoprecipitation-MS analysis and by PLA experiments (Fig. 3a and Supplementary Fig. 12c). MARCH5 is autoubiquitylated and degraded by the proteasome (Supplementary Fig. 12d), and we observed that ablation of MFN2 decreased its protein levels (Supplementary Fig. 12e). This is consistent with previous findings proposing that, by binding to MARCH5, MFN2 sequesters this E3 ligase and inhibits its catalytic activity[72,73]. Nevertheless, MFN2 ablation did not interfere with the interaction of MARCH5 with the proteasome (Supplementary Fig. 12f), suggesting that WT and 2KO cells might have similar levels of MARCH5 ubiquitylation. Together, these findings indicate that MFN2 regulates the stability of a subset of proteins, including MFN1 and PINK1.

## MFN2 binds to cytosolic chaperones and prevents protein aggregation

In addition to ubiquitin regulators, MFN2 also enriched several proteins involved in clearance of protein aggregation (Fig. 3a). For example, the heat-shock-cognate HSC70 from the HSP70 protein family, HSP90, and their adaptor protein STIP1, were identified as MFN2 interactors. In addition, MFN2 binds to the chaperone regulator BAG2, the protein folding accelerator PPIL4 PPIase, the histone chaperone NAP1L1 and the ubiquitin-like protein UBL4A. HSC70 is constitutively expressed and plays major quality control roles, including assisting misfolded

client proteins to refold[74]. Proximity ligation assays with endogenous antibodies confirmed the interaction between MFN2 and HSC70 (Fig. 5a), consistent with recent findings[51], and between MFN2 and BAG2 (Supplementary Fig. 13a). Despite these interactions, MFN2 loss did not alter their protein levels (Supplementary Fig. 13b). Contrarily to what we observed for the proteasome, ubiquitylation inhibition did not affect the interaction of MFN2 with HSC70, suggesting that MFN2 binds the proteasome and HSC70 through independent domains (Fig. 5b). Along this line, absence of MFN2 did not affect the interaction between HSC70 and the proteasome (Supplementary Fig. 13c).

To understand the importance of these interactions and the role of MFN2 in proteostasis, we assessed protein aggregation with the Proteostat Aggresome detection kit. Strikingly, 2KO cells displayed a remarkable accumulation of protein aggregates, which could be res-cued by stable re-expression of MFN2 (Fig. 5c). In contrast, although MFN1 also bound chaperones, ablation of MFN1 did not mimic this effect. The protein aggregates were either distributed throughout the cytoplasm or in proximity to mitochondria, co-staining with the mitochondrial marker TOM20 (Fig. 5d). In conclusion, MFN2, but not MFN1, affects protein aggregation.

## HSC70 and MFN2 cooperate to prevent aggregation of newly synthesized proteins

Biochemical fractionation revealed an accumulation of PINK1 in the urea-resistant insoluble fraction of 2KO cells (Supplementary Fig. 14a). To investigate if protein aggregation could be caused by the protein import defects observed in 2KO cells, we performed Proteostat staining in WT cells treated with the depolarizing agent CCCP, which blocks protein import and increases PINK1 levels (Supplementary Fig. 10d). Surprisingly, however, this significantly reduced protein aggregation (Fig. 5e and Supplementary Fig. 14b), indicating that PINK1 accumulation is not the cause for the aggregation in 2KO cells. Moreover, CCCP treatment did not alter the protein aggregation propensity of 2KO cells. This prompted us to explore if, as suggested to occur in WT cells upon import clogging[75], MFN2 KO could be trig-gering an activation of the integrated stress response (ISR), conse-quently inhibiting protein translation. However, adding ISRIB, which counteracts the ISR and prevents translation inhibition[76] didn't cause protein aggregation of CCCP-treated cells, neither did it affect PINK1 levels (Supplementary Fig. 14c, d). Next, the importance of HSC70 in protein aggregation was investigated. Inhibition of HSC70 by VER-1555008 rendered 2KO cells more susceptible to protein aggregation. In contrast, WT cells were not significantly affected by this treatment, reinforcing the role of MFN2 in proteostasis buffering capacity (Fig. 5f and Supplementary Fig. 14e). As previously reported, VER-1555008 treatment reduced PINK1 levels[77], in both WT and 2KO cells, again pointing to the lack of correlation between PINK1 levels and protein aggregation (Supplementary Fig. 14f). Among other functions, cha-perones from the HSP70 family help to maintain newly translated proteins in a controlled-unfolded state until their translocation into the mitochondria[18,78,79]. Consistently, translation inhibition largely

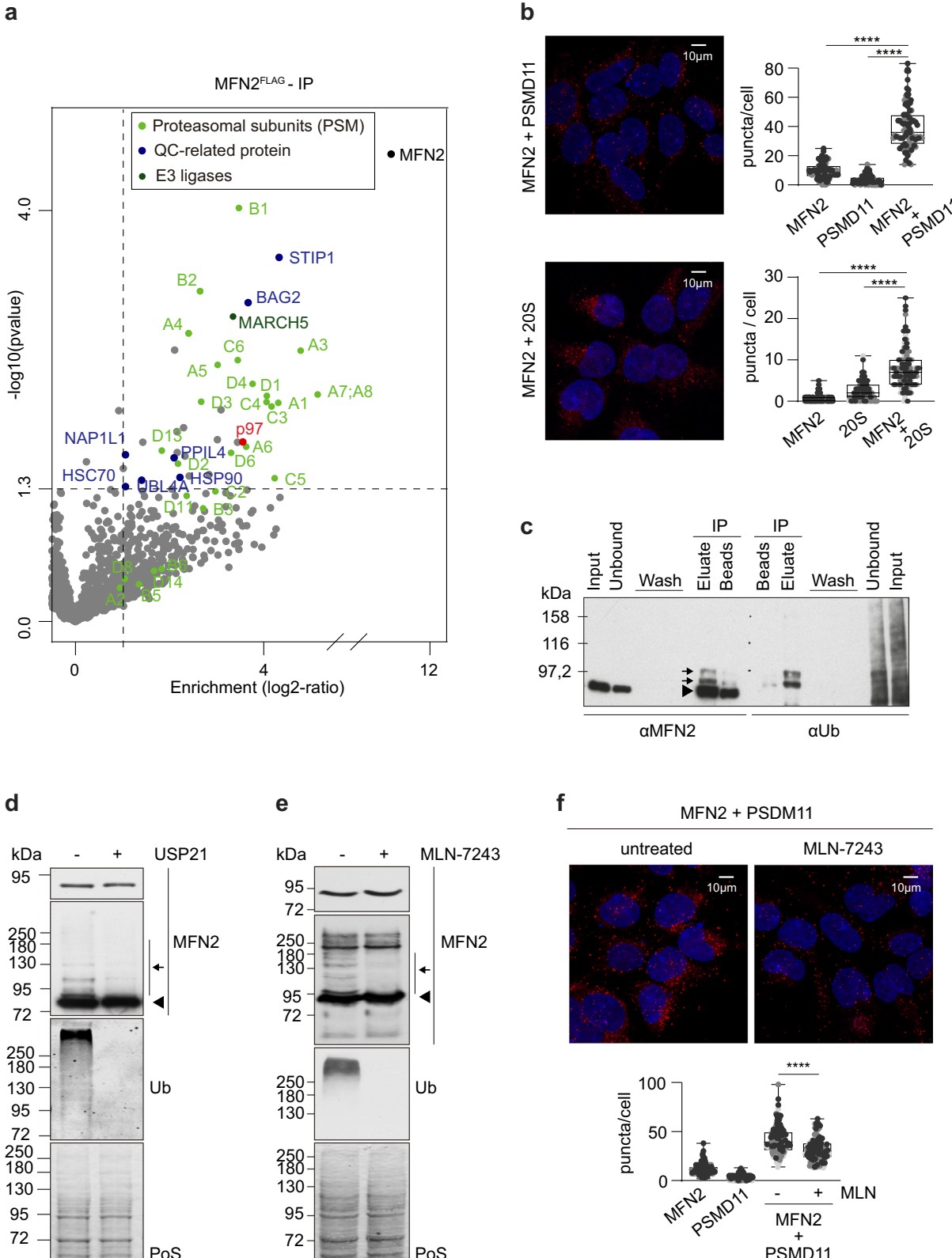

suppressed protein aggregation in 2KO cells (Fig. 5g and Supplementary Fig. 14g). In conclusion, MFN2 interacts with HSC70, assisting it in preventing aggregation of newly synthesized proteins.

## Protein aggregation depends on ubiquitylation

To investigate if the accumulation of protein aggregates in 2KO cells is also due to an impairment in their clearance, for example by aggrephagy, we assessed the importance of the autophagy and proteasomal systems. However, inhibiting autophagy with Bafilomycin A did not cause protein aggregation in WT cells, neither did it alter the aggregation propensity of 2KO cells (Fig. 6a), indicating that clearance of aggregates by autophagy is not impaired in 2KO cells. In contrast, inhibition of the proteasome with MG132 induced protein aggregation (Fig. 6b), supporting the importance of this quality control machinery

**Fig. 3 | MFN2 is constitutively ubiquitylated and binds the proteasome.**
**a** Volcano plot of significantly enriched proteins ($p = 0,05$, 2-fold enrichment) identified by label-free quantification of immunoprecipitated MFN2[FLAG]. Proteasomal subunits: light green. Quality control (QC)-related proteins: blue. E3 ligases: dark green. ($n = 3$ biological replicates). **b** Confocal images of proximity ligation assays of HEK WT cells, with antibodies against MFN2 and PSMD11 (left upper panel) or against MFN2 and 20S subunits (left lower panel) (in red) and DAPI staining (in blue). Scale bar: 10 μm. Quantification of number of puncta per cell (right panels) using exclusively each of the antibodies or both ($n = 2$ biological replicates). Ordinary one-way ANOVA was applied. $P$ values: **** <0.0001. Boxes borders indicate the 25% and 75% quantiles. The minimum and maximum values are shown by the whiskers excluding outliers. **c** Western blot analysis of MFN2, immunoprecipitated from total cell lysates of 1KO cells, immunoblotted with anti-MFN2 and anti-ubiquitin. Arrowhead points to endogenous unmodified MFN2, arrows point to modified MFN2, corresponding to ubiquitylated forms. **d** Western

blot analysis as in (**c**) of total cell lysates from HEK WT cells, untreated (-) or treated (+) with USP21 DUB (30 min, 5 μm, 37 °C), immunoblotted with anti-MFN2 and anti-ubiquitin. Arrowhead points to endogenous unmodified MFN2, arrow points to modified MFN2, corresponding to ubiquitylated forms. **e** Western blot analysis of MFN2 immunoprecipitated from total cell lysates of WT cells, untreated (-) or treated with MLN-7243 (+; 0.5 μM, 4 h), immunoblotted with anti-MFN2 and anti-ubiquitin. Arrowhead points to endogenous unmodified MFN2, arrow points to modified MFN2, corresponding to ubiquitylated forms. Staining of the input with PoS was used as loading control. **f** Proximity ligation assay of HEK WT cells untreated (-) or treated with MLN-7243 (+; 0.5 μM, 4 h) as in (**b**) ($n = 3$ biological replicates). Ordinary one-way ANOVA was applied. $P$ value: **** <0.0001. Boxes borders indicate the 25% and 75% quantiles. The minimum and maximum values are shown by the whiskers excluding outliers. Source data are provided as Source Data file.

---

in preventing aggregation of newly synthesized proteins[80]. Moreover, treatment of 2KO cells with MG132 further aggravated the accumulation of protein aggregates, consistent with a cooperation between MFN2 and the proteasome. We further tested the presence of known aggresomal markers in the 2KO aggregates, namely the centrosomal protein gamma-tubulin[81] and vimentin[82]. Prolonged proteasomal blockage leads to the formation of aggresomes, which result from active transport and coalescence of small aggregates to the centromere, as can be detected by Proteostat staining (Supplementary Fig. 15a, b). While gamma-tubulin colocalized with the Proteostat signal, as expected, no particular enrichment could be found at the smaller aggregates in 2KO cells (Supplementary Fig. 15a). In contrast, vimentin caged both the aggresome and the smaller aggregates, present in WT upon prolonged MG132 treatment or in 2KO cells (Supplementary Fig. 15a). This is consistent with its role in proteotoxic stress and proteasomal recruitment to protein aggregates[82] and further supports the importance of protein turnover in 2KO cells (Fig. 6b). Finally, we investigated how ubiquitylation capacity interferes with protein aggregation. Interestingly, preventing protein ubiquitylation, via inhibition of the ubiquitin activating enzyme UBA1 (E1) with MLN-7243, led to a reduction of aggregates in 2KO cells (Fig. 6c), which is consistent with the importance of ubiquitylation for their formation. In summary, aggregates present in MFN2 KO cells require protein ubiquitylation and are subjected to proteasomal turnover.

## CMT2A patient fibroblasts mimic fusion-independent quality control role of MFN2

Due to the prominent role of MFN2 in mitochondrial fusion, we set out to clarify to which extent protein aggregation correlates with defective mitochondrial morphology. To answer this question, we analyzed how treatments differently altering mitochondrial morphology in WT and 2KO cells correlated with the aggregation propensity. We noticed that VER-1555008 treatment partially corrected the perinuclear clustering and swelling properties of mitochondria in 2KO cells (Fig. 7a). However, it significantly increased protein aggregation (Fig. 5f). In contrast, mitochondrial fragmentation caused by MFN1 depletion had no effect on protein aggregation (Fig. 5c). Finally, treatment of WT cells with CCCP, which also caused mitochondrial fragmentation (Fig. 7b), reduced protein aggregation (Fig. 5e). In turn, CHX treatment reduced protein aggregation in 2KO cells (Fig. 5g), yet it did not alter their mitochondrial morphology (Fig. 7c). In WT cells, CHX treatment induced the expected hyper-elongation of mitochondria[83], but didn't significantly alter protein aggregation. In sum, similar alterations in mitochondrial morphology can have different outcomes in protein aggregation, and vice-versa.

In order to evaluate the physiological relevance of our findings, we studied protein aggregation in human primary skin fibroblasts of three CMT2A patients carrying different MFN2 point mutations – R94Q, R94W and R104W. Mitochondrial morphology was maintained

from controls to CMT2A patient fibroblasts (Fig. 7d), as reported[84,85]. Interestingly, contrarily to the two healthy individuals (Control A and B), CMT2A patient fibroblasts presented augmented protein aggregation (Fig. 7d). Importantly, this also indicated that protein aggregation does not impinge in the fusogenic capacity of mutant MFN2. Finally, full-length PINK1 was not increased in these CMT2A primary fibroblasts (Supplementary Fig. 16), further supporting the lack of correlation between membrane potential and protein aggregation. This is in agreement with the results observed in human cell lines stably expressing the CMT2A K357N variant (Fig. 2f and Supplementary Fig. 6b). In conclusion, MFN2 disease-associated variants recapitulate the proteostasis defect of 2KO cells.

## Discussion

Our study unravels MFN2 as a component required for constitutive surveillance of protein quality control (Fig. 8). Cellular proteostasis relies on cytosolic chaperones and on the proteasome apparatus and mitochondrial functionality is particularly dependent on both machineries. Strikingly, MFN2 physically interacts with both. Together with these major housekeeping systems, MFN2 prevents the accumulation of protein aggregates, a novel MFN2-linked phenotype that we found replicated in a disease context. MFN2 also controls MFN1 degradation by the proteasome, whereas MFN1 ablation did not affect MFN2 nor protein aggregation. In addition, MFN2 KO impairs protein import capacity, which consequently triggers the accumulation of the PINK1 precursor and upregulation of mitophagy. Therefore, despite the high similarity between MFN1 and MFN2 and their unquestionable shared capacity to mediate mitochondrial fusion, we describe a unique role of MFN2 in maintaining protein homeostasis. Moreover, by revealing a function for MFN2 beyond promoting membrane fusion, our study adds a new layer to its previously acknowledged non-canonical roles and simultaneously strengthens the impact of mitochondrial dynamics components in the maintenance of cellular homeostasis[2,34].

The threat of impaired mitochondrial functionality for cellular proteostasis is becoming increasingly acknowledged, since it not only affects the unimported mitochondrial precursor proteins, but also promotes a general increase in cytosolic protein aggregation[8–12,25–28]. Given that most of the mitochondrial proteome is genomically encoded and translated by cytoplasmic ribosomes, newly synthesized mitochondrial proteins need to be tightly controlled. On one hand, the cytosolic chaperones HSC70 and HSP90 grant protection of newly translated proteins from misfolding and aggregation, maintaining them in a controlled-unfolded state and safely routing them until their delivery to mitochondria. On the other hand, the proteasome controls the turnover of faulty translated or misfolded proteins, also preventing their aggregation. Remarkably, MFN2 binds to the main cellular chaperones HSC70 and HSP90, together with their co-factor STIP1, as well as the 19 and 20S proteasome complexes. These interactions, which are specific to MFN2 and not to MFN1, agree with high throughput

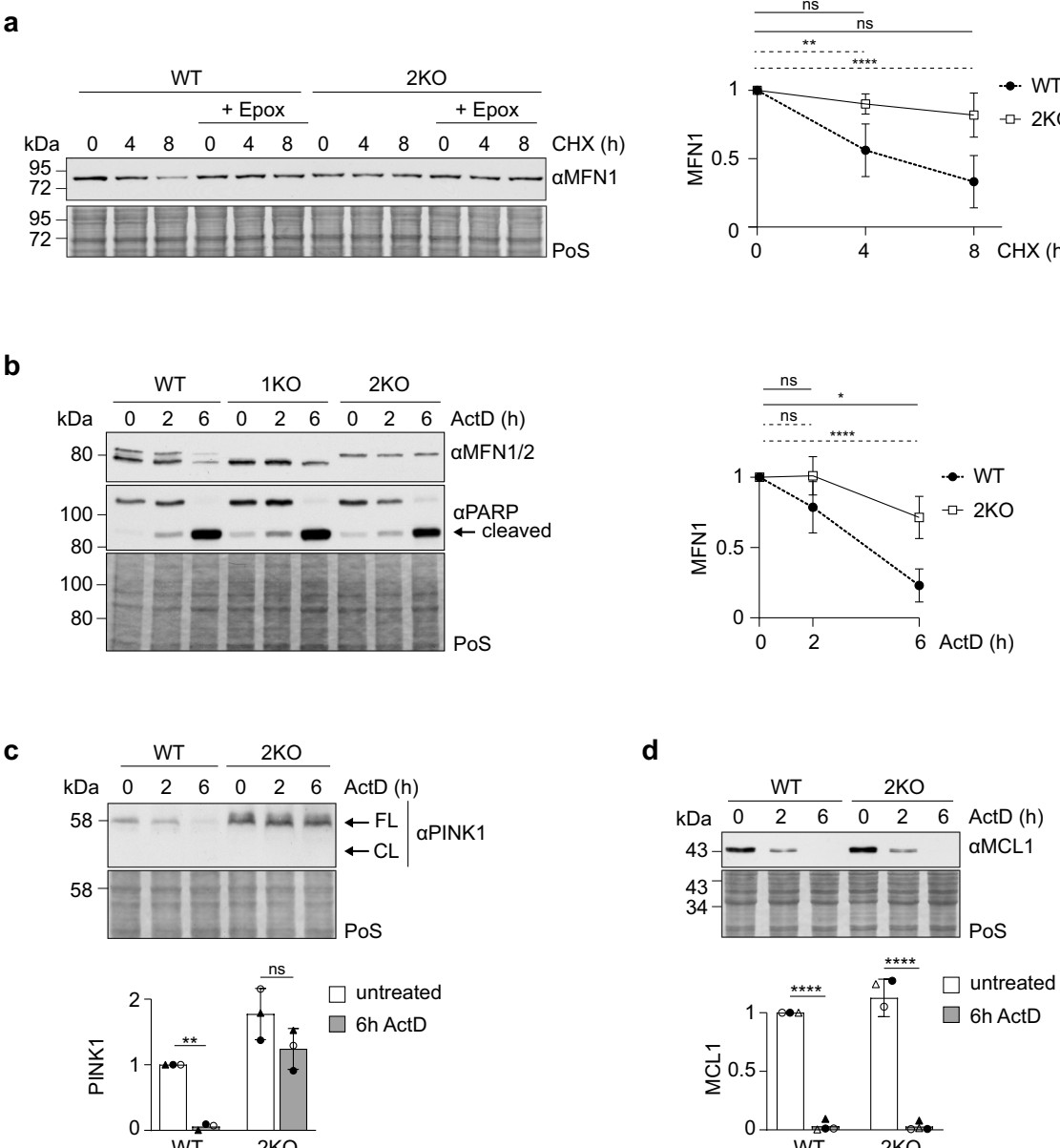

**Fig. 4 | MFN2 controls MFN1 turnover. a** Western blot analysis of total cell lysates from HEK WT cells untreated (0) or treated with cycloheximide (CHX, 100 μg/mL) for 4 or 8 hr, simultaneously treated with epoxomicin (1 μM), when indicated, and immunoblotted with anti-MFN1. Staining of total protein with PoS was used as loading control. Shown are the mean and SD of the individual timepoints, relative to the correspondent untreated control, labeled as "0" on the Western blot (*n* = 3 biological replicates). Two-way ANOVA was applied. P L-R: ** 0.0023, **** <0.0001. **b** Western blot analysis as in a) of total cell extracts from HeLa WT, 1KO and 2KO cells untreated (0) or treated for 2 or 6 h with ActD (1 μM), immunoblotted with anti-MFN1/2 and anti-PARP1. Shown are the mean and SD of the individual time-points, relative to the correspondent untreated control, labeled as "0" on the Western blot (*n* = 4 biological replicates). Two-way ANOVA was applied. P L-R: *

0.0367, **** <0.0001. **c** Western blot analysis as in a) of total cell extracts from HeLa WT and 2KO cells untreated (0) or treated for 2 or 6 hr with ActD (1 μM), immunoblotted with anti-PINK1 (FL: full-length; CL: cleaved). Bars represent the mean fold change relative to WT ± SD (*n* = 3 biological replicates). Individual values of each experiment are discriminated in triangles and white or black filled circles. Two-way ANOVA was applied. *P* value: * 0.0076. **d** Western blot analysis as in a) of total cell extracts from HeLa WT and 2KO cells untreated (0) or treated for 2 or 6 h with ActD (1 μM), immunoblotted with anti-MCL1. Bars represent the mean fold change relative to WT ± SD (*n* = 3 biological replicates). Individual values of each experiment are discriminated in triangles and white or black filled circles. Two-way ANOVA was applied. *P* value: **** <0.0001. Source data are provided as Source Data file.

data[86]. Strikingly, absence of MFN2 causes a general increase in protein aggregation. Moreover, inhibition of protein translation prevents aggregate accumulation in MFN2 KO cells, underlining the importance of MFN2 in newly synthesized polypeptides, as it is the case for HSC70 and for the proteasome. Finally, aggregation propensity is much increased upon simultaneous inhibition of MFN2 and HSC70, or MFN2 and the proteasome, further consistent with a cooperative role of MFN2 with HSC70 and the proteasome. As expected, MFN2 is

constitutively ubiquitylated and binding to the proteasome is depen-dent on it, supporting a direct effect of MFN2 in quality control. Finally, inhibition of the E1 ubiquitin-activating enzyme UBA1 attenuated the aggregation propensity of MFN2 KO cells, consistent with the impor-tance of protein ubiquitylation. These data suggest that MFN2 assists the chaperone network in preventing aggregation of newly synthesized proteins and assists the proteasome in promoting their turnover.

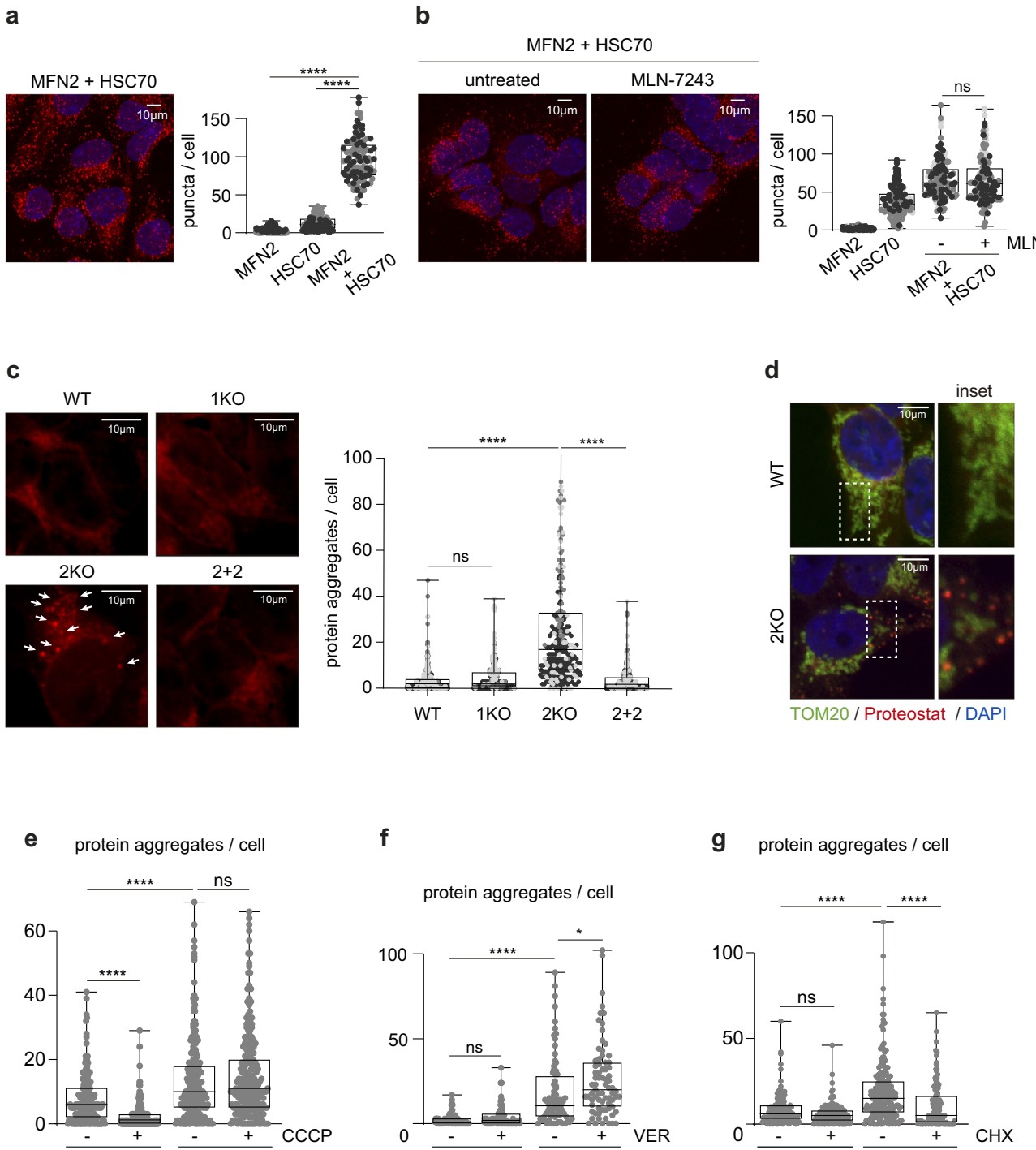

**Fig. 5 | Absence of MFN2 causes aggregation. a** Confocal images of proximity ligation assay of HEK WT cells, with antibodies against MFN2 and HSC70 (in red) and DAPI staining (in blue) (left panel). Scale bar: 10 μm. Quantification of number of puncta per cell (right panel) using exclusively each of the antibodies or both (*n* = 2 biological replicates). Ordinary one-way ANOVA was applied. *P* value: **** <0.0001. Boxes borders indicate the 25% and 75% quantiles. The minimum and maximum values are shown by the whiskers excluding outliers. **b** Proximity ligation assay of HEK WT cells untreated (-) or treated with MLN-7243 ( + ; 0.5 μM, 4 h), as in a) (*n* = 3 biological replicates). Ordinary one-way ANOVA was applied. Boxes borders indicate the 25% and 75% quantiles. The minimum and maximum values are shown by the whiskers excluding outliers. **c** Staining of protein aggregation with the PROTEOSTAT® Aggresome detection kit (in red) in HEK WT, 1KO, 2KO and 2 + 2 cells (left panel). Scale bar: 10 μm. Quantification of number of protein aggregates per cell (right panel). *n* = 3 biological replicates, color-coded in greyscale, with at

least 100 cells in each were quantified. Ordinary one-way ANOVA was applied. *P* value: **** <0.0001. Boxes borders indicate the 25% and 75% quantiles. The minimum and maximum values are shown by the whiskers excluding outliers. **d** Confocal images of HEK WT and 2KO cells co-stained with PROTEOSTAT® Aggresome detection kit (in red), anti-TOM20 (in green) and DAPI (in blue). Scale bar: 10 μm. Insets of white dotted boxes are shown on the right side of each image. **e**–**g** Quantification of amount of protein aggregates per cell in HEK WT and 2KO cells, untreated (-) or treated (+) with CCCP (20 μM, 2 h) **on e**, with VER-155008 (50 μM, 2 h) **on f**, or with CHX (5 μM, 5 h) **on g**. *n* = 1 biological replicate with at least 100 cells was quantified. Ordinary one-way ANOVA was applied. *P* value: **** <0.0001 (on e), P L-R: **** <0.0001, * 0,0166 (on f), *P* value: **** <0.0001 (on g). Boxes borders indicate the 25% and 75% quantiles. The minimum and maximum values are shown by the whiskers excluding outliers. Source data are provided as Source Data file.

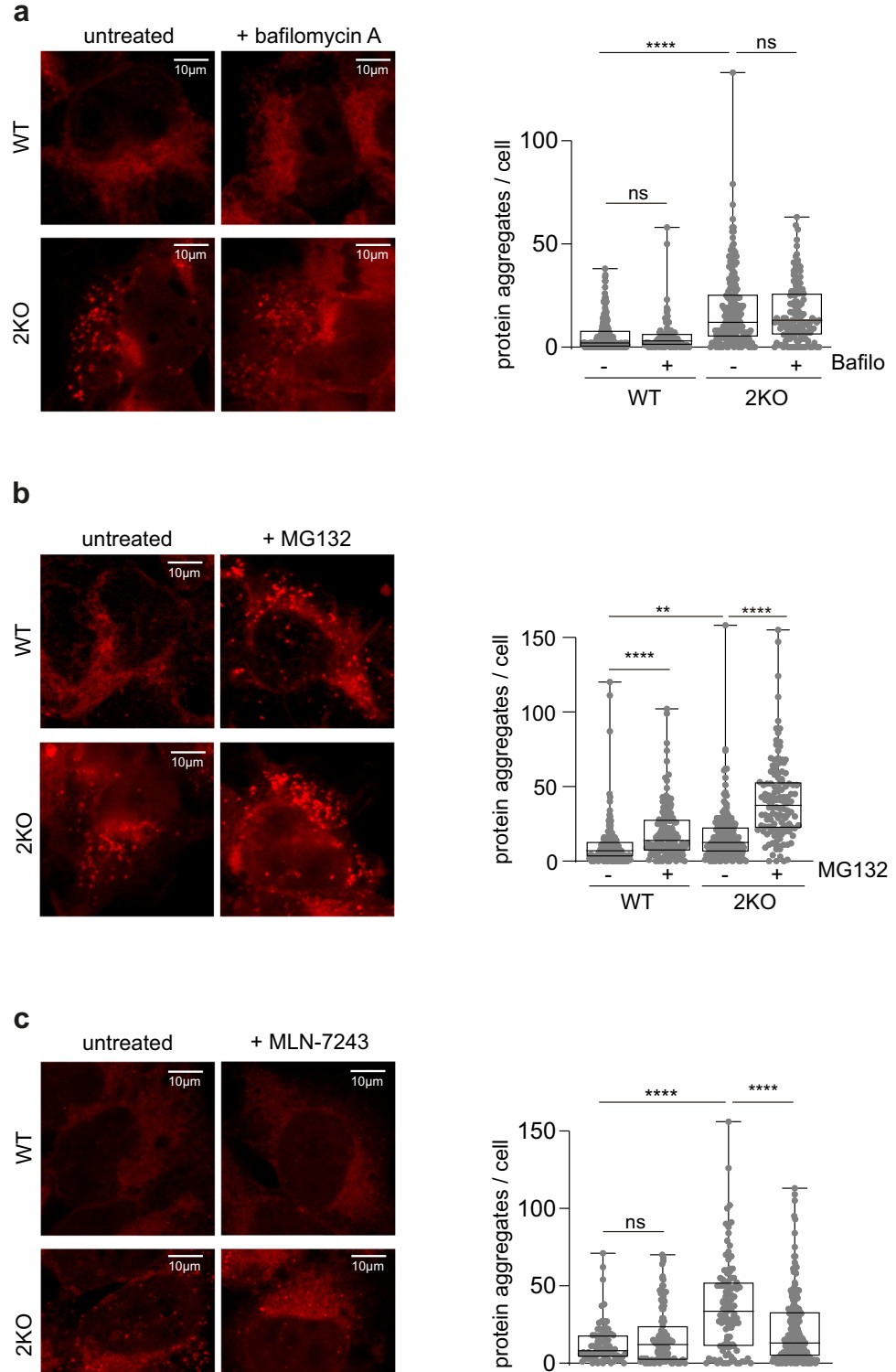

**Fig. 6 | Protein aggregation in 2KO cells depends on ubiquitylation. a–c** Right panels: Confocal images of HEK WT and 2KO cells, untreated or treated with Bafilomycin A (100 nM, 4 h) **on a**, with MG132 (10 μM, 5 h) **on b**, or with MLN-7243 (MLN; 0.5 μM, 4 h) **on c**, stained with PROTEOSTAT® Aggresome detection kit. Scale bar: 10 μm. Left panels: Quantification of number of protein aggregates per cell in HEK WT and 2KO cells for the conditions depicted in (**a–c**). n=1 biological replicate with at least 100 cells was quantified. Ordinary one-way ANOVA was applied. *P* value: **** <0.0001 (on **a**), P L-R: **** <0.0001, ** 0.003, **** <0.0001 (on **b**), *P* value: **** <0.0001. Boxes borders indicate the 25% and 75% quantiles. The min and maximum values are shown by the whiskers excluding outliers. Source data are provided as Source Data file.

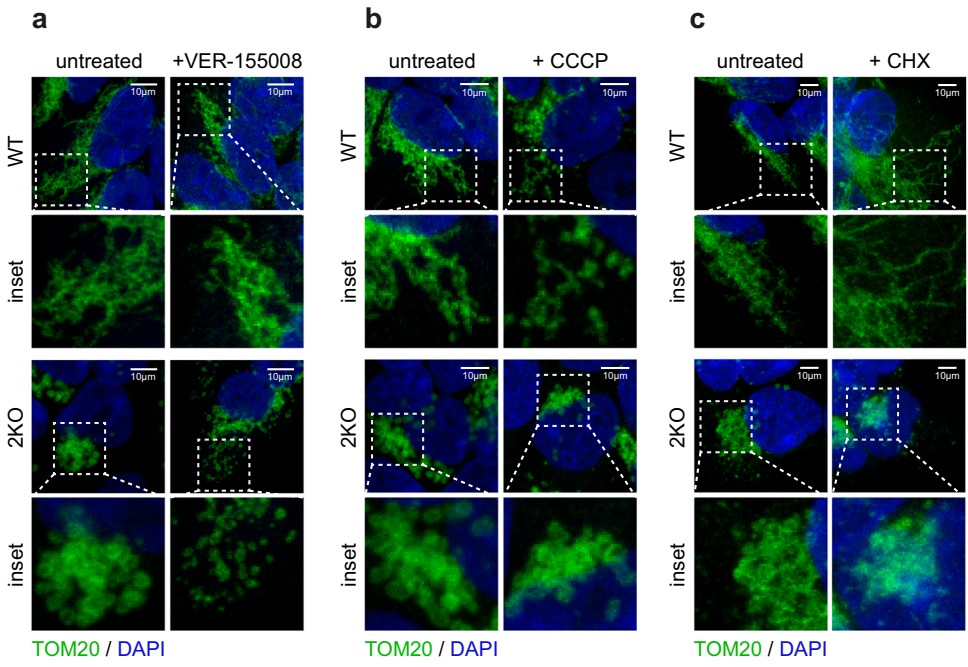

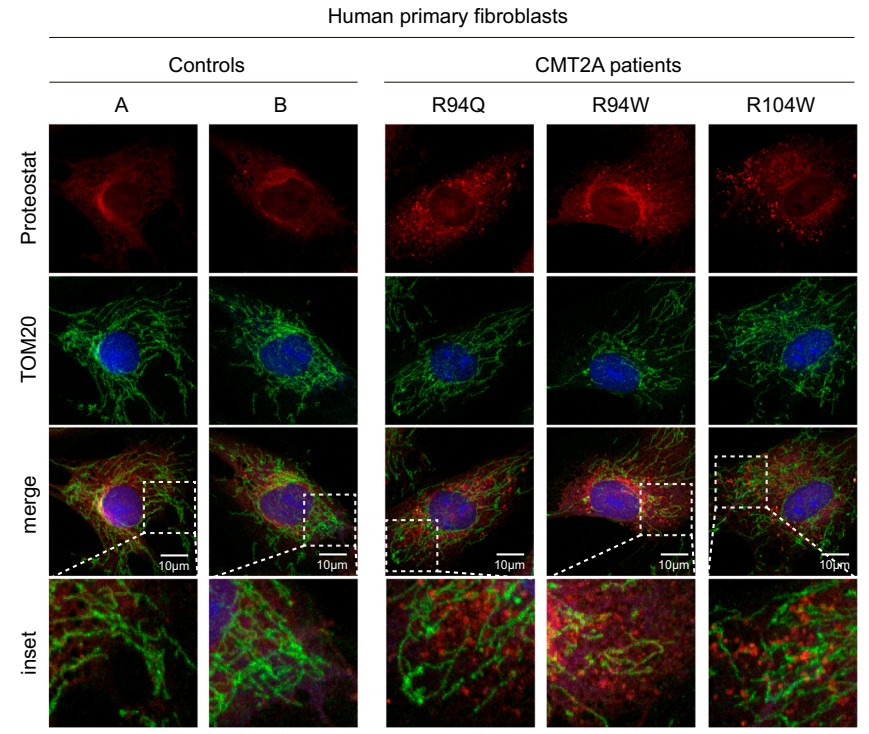

**Fig. 7 | CMT2A patient fibroblasts have increased protein aggregation.**
**a–c** Confocal images after immunostaining with anti-TOM20 (in green) and DAPI (in blue), of HEK WT and 2KO cells, untreated or treated with VER-155008 (50 μM, 2 h) **on a**, with CCCP (20 μM, 2 h) **on b**, or with CHX (5 μM, 5 h) **on c**. Scale bar: 10 μm. Insets of white dotted boxes are shown below each image. **d** Confocal images of two control and three CMT2A patients' primary fibroblasts carrying the point mutations R94Q, R94W and R104W in MFN2, co-stained with the PROTEOSTAT® Aggresome detection kit (in red), anti-TOM20 (in green) and DAPI (in blue). Scale bar: 10 μm. Insets of white dotted boxes are shown below each image.

MFN2 expression was sufficient to rescue mitochondrial morphology of both single and double MFN1 and MFN2 knockouts. In contrast, MFN1 expression could only rescue MFN1 KO cells. These observations underline a broader importance of MFN2 in determining mitochondrial morphology and ultrastructure in human cells. In turn, MFN1 has been more prominently reported to act in stress-induced hyperfusion events[83]. Beyond mitochondrial fusion defects, depletion of MFN2 impaired cellular proliferation and impacted the cellular proteome, leading to a marked mitochondrial mass reduction and to alterations in the levels of more than 1400 proteins. In MFN2 KO cells, OXPHOS subunits and mitochondrial ribosome components were downregulated, along with impaired oxidative phosphorylation and oxygen consumption rates. This is consistent with previous studies[35,38,39,44,45] and agrees with the deeply affected mitochondrial

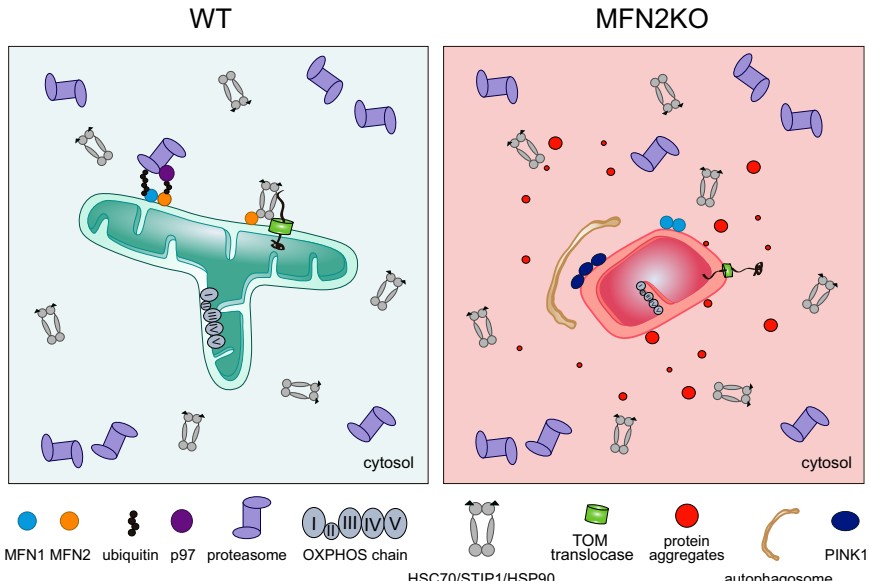

**Fig. 8 | Roles of MFN2 in proteostasis control.** MFN2 maintains healthy mitochondria, which present in a tubular form, and exhibit functional oxidative phosphorylation (OXPHOS) capacity. This provides a healthy cellular environment, with active mitochondrial import, surveyed by cytosolic chaperones that bind MFN2. Furthermore, MFN2 is constitutively modified with short ubiquitin chains, binds p97 and the proteasome, allowing the turnover of other outer membrane proteins, including its cognate MFN1 and PINK1. In contrast, cells depleted for MFN2 (MFN2KO) fail to locally maintain this quality control at the mitochondrial surface. Mitochondria are swollen, fragmented and dysfunctional, with extremely reduced OXPHOS capacity. TOM translocase components are also decreased, reducing import capacity into mitochondria. Consequently, PINK1 accumulates, causing the upregulation of mitophagy, and proteins abnormally aggregate outside mitochondria.

functionality upon MFN2 ablation[38]. Additionally, fatty acid oxidation and ROS/glutathione pathways were upregulated in MFN2 KO cells, as observed in OXPHOS and Complex V deficiency models[87,88].

In parallel with mitochondrial dysfunctionality, MFN2 depletion was accompanied by downregulation of import components, in both human cell lines and a neuronal mouse cell model. In particular, the TOM receptor components TOM20 and TOM70, responsible for the recognition of mitochondrial precursor proteins before their import, and the TOM40 import channel component, were all decreased. The consequent alterations in PINK1, a well-known marker of mitochondrial protein import defects[63], confirmed the physiologic impact of low TOM levels. Indeed, loss of MFN2 resulted in an accumulation of its long form under basal conditions and a decrease of its short form under proteasomal blockage, both consistent with defective protein import into mitochondria in MFN2 KO cells. Corroborating the relevance of MFN2 in protein survey and in the regulation of PINK1 levels, MFN2 KO cells exhibited persistent lipidation of the autophagy adaptor LC3 under mitophagy induction conditions, as well as mildly increased mitophagy levels. Importantly, the defects caused by loss of MFN2 could not be mimicked by ablation of MFN1, confirming unique roles of MFN2 in human cells.

The phenotypes of MFN2 KO cells, including mitochondrial dysfunction, poor OXPHOS function, defective protein import and mitophagy, are known to induce cytosolic protein aggregation and cellular stress responses[5,11,30]. Mitochondria and the proteasome can cooperate in the resolution of cytosolic proteotoxic stress[5,29]. This includes upregulation of the mitochondrial unfolded protein response (mtUPR), to induce mitochondrial proteolytic capacity[17], or a role of FUNDC1 and HSC70 to promote import of unfolded cytosolic proteins into mitochondria[18]. In yeast, the ER surface-mediated targeting pathway (ER-SURF), including the J-protein Djp1, allows to keep proteins mistargeted to the ER membrane in an import-competent state and transports them back to mitochondria[89], a pathway possibly present in mammals[24]. In turn, ubiquitin-dependent pathways surveying mitochondrial protein import have been identified in yeast, such as the

mitochondrial compromised protein import response (mitoCPR)[22] or the mitochondrial protein translocation-associated degradation (mitoTAD)[23]. In mammals, a mechanism similar to mitoCPR has just been reported[75] and, reminiscent of mitoTAD, MARCH5 was proposed to constitutively regulate mitochondrial import together with the DUB USP30[90]. In the event of cytosolic accumulation of non-imported precursors, which causes protein aggregation[25,26], pathways relying on upregulation of the proteasome and the chaperone network were also identified, as for example the unfolded protein response activated by mistargeting of proteins (UPRam)[20] or the mitochondrial precursor over-accumulation stress (mPOS)[21]. These pathways were nevertheless not upregulated in our model, supporting that MFN2 controls a different cellular stress response from the ones previously identified.

Strikingly, we observed that primary fibroblasts from CMT2A patients mimicked the propensity for accumulation of protein aggregates observed in MFN2 KO cells, which allows devising new strategies for this incurable disease[57,58]. Beyond CMT2A, several neurodegenerative pathologies characterized by protein aggregation show less expression of MFN2[52,53,91,92], consistent with a causative role of MFN2 in protein aggregation. Moreover, other mitochondrial morphology proteins have been linked with neuronal protein aggregation in diseases such as Alzheimer's and Huntington's[54,93]. For example, DRP1 inhibition partially reduces aggregation, both in a hereditary spastic paraplegia neuronal model and in a Huntington's disease cellular model[94,95]. Our findings allowed to assess if the aggregation propensity depends on their roles in membrane remodeling or if, instead, reflects non-canonical functions. Notably, protein aggregation propensity was clearly not a direct consequence of mitochondrial shape alterations. On one hand, MFN1 KO cells did not display signs of protein quality control defects, suggesting no strict correlation with fragmented mitochondria. Consistently, mitochondrial fragmentation caused by depolarization with CCCP did not lead to protein aggregation. On the other hand, induction of mitochondrial hypertubulation had no effect on protein aggregation. Furthermore, attenuation of the defective mitochondrial network of MFN2 KO cells, which occurred upon VER-

1555008 treatment, did not attenuate but rather increased their aggregation propensity. Finally, the CMT2A mutant cells analyzed, which accumulate aggregates, presented tubular mitochondria. Therefore, besides corroborating the lack of correlation between mitochondrial morphology and protein aggregation, our findings reveal a remarkable physiological relevance of MFN2 in proteostasis and disease.

In conclusion, the present study reveals a previously unknown role of MFN2 in the regulation of protein quality control, by preventing proteostasis imbalance and cytosolic protein aggregation. Interestingly, this surveillance function is not shared by its homologue MFN1 and is independent of mitochondrial fusion. Altogether, the effects of MFN2 depletion are consistent with a greater pathophysiological relevance of MFN2 than of MFN1, exemplified by the involvement of MFN2 in metabolic disorders and neurodegeneration. In addition, a protective role of MFN2 against non-alcoholic fatty liver disease (NAFLD) was recently described, involving both the MFN2 full length protein as well as shorter forms that result from alternative *Mfn2* splice variants[48,49]. Along the same line, the importance of MFN2 levels and MFN2-HSC70 interaction in myocardial lipotoxicity has just been reported[51]. Therefore, beyond its potential roles in CMT2A, the newly-described function of MFN2 as a proteostasis gatekeeper also provides hints for therapeutical strategies in diseases associated with low levels of MFN2.

## Methods

### Cell culture and treatments

All HEK293 (Invitrogen, R75007) (from here on referred to as HEK) and HeLa cells (ATCC, CCL-2) used in this study were maintained in DMEM-GlutaMAX containing 4,5 g/L glucose and supplemented with 10% fetal bovine serum (FBS), 100 μM non-essential aminoacids, 1 mM sodium pyruvate and 10% penicillin/streptomycin.

Knockout of MFN1 and/or MFN2 was performed by a double nickase CRISPR-Cas9 strategy, targeting the N-terminus in exon 4 of MFN1 and exon 5 of MFN2, with paired single guide RNAs (gRNA). For this, $4 \times 10^5$ cells were plated per well of a 6-well plate and transiently transfected with the plasmid pX335 encoding Cas9 nickase targeted against MFN1 or MFN2 by two different guide RNAs (constructs 1 and 2 and constructs 5 and 6, respectively, Supplementary Table 1). After transfection for five consecutive days, cells were diluted into single cell on 96-well plates and screened by Western blot for knockout candidates. Furthermore, the homozygous knockouts were confirmed by sequencing.

Stable Flp-In T-Rex HEK293 expressing MFN2[FLAG] or the mutant variant K357N were generated using the HEK MFN2 KO cell lines. For this, $6 \times 10^5$ cells were plated per well of a 6-well plate and co-transfected with the pcDNA5 plasmid encoding MFN2-3xFLAG or MFN2-K357N-3xFLAG, respectively, and the pOG44 Flp-Recombinase Expression Vector for constitutive vector expression (Supplementary Table 1) in a ratio of 1:4, respectively. Transfection was performed as mentioned above. 24 h post-transfection, cells were expanded into three 15 cm dishes and for 10 consecutive days, were selected with hygromycin B-containing medium (100 μg/ml). After selection, single clones were picked and expanded. A screening of the clones was performed via Western blotting and assessment of MFN2 and FLAG protein levels. The selected clones were confirmed by sequencing. When mentioned, MFN1[FLAG] expression was induced by tetracycline, for 18 h, at the concentrations indicated in the figures.

All chemicals used for cell treatments were prepared in DMSO and the treatments were performed by medium change containing the respective chemical at the final concentration and for the time indicated in the figure legends.

### Transient transfection

For transient transfection, cells were transfected one day after seeding with 1.5 μg of the indicated plasmid DNA (Supplementary Table 1) with GeneJuice® transfection reagent in Opti-MEM™ reduced serum medium and collected after 24–48 h.

### Downregulation using siRNA

For downregulation of MFN1 or MFN2, cells were transfected one day after seeding with 50 pmol of the desired siRNA (Supplementary Table 2) with Lipofectamine™ RNAiMAX in Opti-MEM™ reduced serum medium and collected 72 h later.

### Primary human fibroblasts

Primary human fibroblasts were obtained from skin biopsy from 5 individuals:2 healthy patients, both male and with 46 and 55 years old. 3 patients with Charcot-Marie-Tooth Type 2 A: R94Q - female 60 years old; R94W - female 43 years old; R104W - male 30 years old. Written informed consent was obtained from all participants (Ethics Committee from the Angers University Hospital approval: CPP Ouest 6 – Angers, France; Identification number: CPP1402 HPS2.; Declaration number: 21.04.27.3982; Authorization number: 2021-A00837-34). Patients were sampled during a neurological consultation by a clinical neurologist in CHU Angers. A confirmed pathogenic mutation was required. No patient was remunerated during this study. The witnesses were granted a travel allowance of 30 euros. Sex determination was indicated in the file by the patient's declaration. Gender was not considered in this study since, to date, no clinical data have established a link between the CMT2A phenotype and gender. The gender of the samples was not taken into account. However, we used a balanced mixture of male and female fibroblasts for quantitative western-blot measurements. Primary human fibroblasts were cultured in medium containing 1% sodium-pyruvate, 1% uridine 5 mg/mL, 1% penicilin-streptomycin, 10% decomplemented fetal calf serum, 30% complemented AmnioMax and DMEM F12. To avoid bias due to fibroblast mutations, only cells with a passage of less than 25 were studied.

### Mouse models and Neuronal cultures

Mus Musculus C57BL/6 N mice with loxP-flanked Mfn2 genes were previously described[39]. Mice were housed in groups of up to 5 animals per cage supplied with standard pellet food and water ad libitum with a 12 h light/dark cycle, while temperature was controlled to 21–22 °C. All animal procedures were conducted in accordance with the European, national, and institutional guidelines and were approved by the Landesamt für Natur, Umwelt und Verbraucherschutz, Nordrhein-Westfalen, Germany. Animal work also followed the guidelines of the Federation of European Laboratory Animal Science Associations.

Primary cultures of mouse cortical neurons were prepared as described in ref. [35], with minor modifications. Briefly, cortices obtained from 5–7 E14 embryos of both sexes per litter from at least 3 pregnant females were dissected in HBSS supplemented with 10 mM HEPES and dissociated by means of enzymatic digestion by incubating the tissue in DMEM containing 20 U/ml papain and 1 μg/ml cystein for 20 min at 37 °C, followed by mechanical trituration in DMEM supplemented with 10% fetal bovine serum, obtaining neurons in suspension. These neurons were then seeded at a concentration of $8 \times 10^5$ cells per well of a 6-well plate. To induce recombination in vitro, neurons were treated at Day in vitro (DIV) 2 with 1,5 μL of the following AAV9 viral vectors: AAV9.CMV.-PI.eGFP.WPRE.bGH (Addgene, catalog no. 105530-AAV9) and AAV9.CMV.HI.eGFP-Cre.WPRE. SV40 (Addgene, catalog no. 105545-AAV9). Neurons were treated with AraC 1 μM at DIV3 and again at DIV7 with the semi-feeding. Finally, neurons were collected at DIV11. For protein extraction, primary neurons were scraped in PBS, pelleted at 2400 *g* for 5 min at 4 °C and lysed in a 0.5% Triton X-100, 0.5% sodium deoxycholate and PBS lysis buffer, with proteinase inhibitors. Samples were quantified via Bio-Rad Protein Assay Dye Reagent Concentrate and prepared using a 4x Laemli Sample buffer enriched with b-mercaptoethanol and boiled at 95 °C for 5 min before loading.

## Western blotting

HEK cells, HeLa cells or human primary fibroblasts were harvested with ice-cold phosphate-buffered saline (PBS) and centrifuged at $1000\,g$ at 4 °C for 5 min and washed one again with ice-cold PBS. The cell pellets were resuspended in RIPA lysis buffer (1% Triton X-100, 0,1% SDS, 0,5% sodium deoxycholate, 1 mM ETDA, 50 mM tris pH7,4, 150 mM NaCl and freshly added cOmplete EDTA free protease inhibitor) and shaken at 200 g at 4 °C for 45 min. After lysis, the samples were centrifuged at $16800\,g$ at 4 °C for 15 min and the supernatant was collected. Protein quantification was performed by using Bradford protein assay (Bio-Rad) according to the manufacturer's protocol and 150 μg of protein were denatured at $200\,g$ at 45°C for 20 min in 6x Laemmli Buffer (1 M tris-HCL pH 6,8, 60% glycerol, 4% SDS, 0,1% bromophenol blue and freshly added dithiothreitol (DTT)).

After SDS-PAGE, proteins were blotted into nitrocellulose or polyvinylidene difluoride membranes and protein loading was assessed by Ponceau S staining. The membranes were blocked in 5% non-fat milk in TBS for 1 h. The membranes were incubated overnight in primary antibodies in the concentrations indicated in Supplementary Table 3 and afterwards washed 3 times in TBS and incubated 1 h with the respective HRP-conjugated secondary antibodies in 5% non-fat milk in TBS. After 3 times washing with TBS, the membranes were developed with chemiluminescent detection using selfmade ECL (100 mM tris pH8,5, 0,44% luminol, 0,009% p-coumaric acid, 0,018% $H_2O_2$), Advansta WesternBright® ECL, Amersham™ ECL or Super-Signal™ West Femto ECL and a URIX60 film processor. For enrichment of low-abundant ubiquitylated forms of MFN2 we performed WB as previously described[96].

Unprocessed and uncropped scans of all blots are provided in the Source Data file.

## Immunostaining

For immunostaining, cells were seeded in poly-L-lysine-coated coverslips, were fixed in 4% paraformaldehyde in growth media for 15 min at 37 °C and the fixative solution was washed once with PBS. After fixation the cells were permeabilized with 0,15% Triton X-100 in PBS (0,2% Triton X-100 for fibroblasts) for 15 min and blocked with 2% BSA in PBS for 1 h. The coverslips were incubated with the indicated primary antibodies (Supplementary Table 3) for 1 h, followed by two 20 min washes with PBS. Decoration with the respective fluorescent secondary antibodies and DAPI for nuclear staining was performed for 1 h, followed by two 20 min washes with PBS. The coverslips were mounted onto glass slides with ProLong™ Gold Antifade mounting solution. All microscopy images were acquired with one of the following confocal microscopes: UltraView Vox (Perkin Elmer), Stellaris 5 (Leica) and TCS SP8 gSTED (Leica).

## Protein aggregation staining

For detection of protein aggregates, the PROTEOSTAT® Aggresome detection kit (Enzo) was used. For this, cells were seeded in poly-L-lysine-coated coverslips, fixed in 4% paraformaldehyde in growth media for 15 min at 37 °C and the fixative solution was washed once with PBS. After fixation, the cells were permeabilized with 0,015% Triton X-100 in PBS. cells were fixed and permeabilized as previously described. After washing once with PBS, the coverslips were then incubated with the PROTEOSTAT® Aggresome detection reagent diluted in 1X Assay Buffer (1:1300 dilution) for 30 min at RT. After incubation, the coverslips were washed twice with PBS and mounted onto glass slides as previously described.

In case of co-staining with antibodies, the immunostaining protocol (above) took place as described. After washing the secondary antibody, the coverslips were incubated with the PROTEOSTAT® Aggresome detection reagent diluted in 1X Assay Buffer (1:1300 dilution) for 30 min at RT, washed twice with PBS and mounted onto glass slides as previously described.

Quantification of protein aggregates was performed using Imaris 10.2.0 Image Analysis Software (Oxford Instruments, Andor, Belfast, United Kingdom). The Imaris 10.2.0 cell module was subsequently used to calculate the number of PROTEOSTAT® positive foci per cell, by first defining fluorescence intensity thresholds for DAPI, PROTEO-STAT® foci and the cell followed, by a segmentation in a batch format.

## Electron microscopy

For electron microscopy, both HEK or HeLa cells were grown in Aclar foil coverslips and at a confluency of ~40% the cells were fixed for 30 min at room temperature (RT) followed by 30 min at 4°C with fixation buffer (2% glutaraldehyde, 2,5% sucrose, 3 mM CaCl2 and 100 mM HEPES, pH 7,4). After 3x washing with 0,1 M sodium cacodylate buffer, the cells were incubated with 1% osmiumtetroxide, 1,25% sucrose, 1% potassium ferricyanide in 0,1 M sodium cacodylate buffer for 1 h at 4°C and again washed 3x with 0,1 M sodium cacodylate buffer. This was followed by a series of incubations in EtOH (50%, 70%, 90% and 100%) for 7 min at 4°C. Two EPON/EtOH incubations were followed (1:1 for 1 h and 3:1 for 2 h) at 4°C. Finally, the cells were embedded in EPON buffer for 72 h at 62 °C, cut in 70 nm sections on an ultramicotome (UC6, Leica) and contrasted with 1,5% uranylacetate aqueous solution for 15 min at 37 °C. Afterwards, the cells were washed 5x in water and incubated 4 min in lead citrate, washed again 5x in water and dried on a filter paper. Imaging was performed with a transmission electron microscope (JEM 2100 Plus, JEOL), a OneView 4 K camera (Gatan) at 80 kV at room temperature.

## Immunoprecipitation of MFN1$^{FLAG}$ or MFN2$^{FLAG}$

1KO or 2KO cells stably transfected with MFN1$^{FLAG}$ or MFN2$^{FLAG}$, respectively, were harvested with ice-cold PBS as previously described and lysed with Triton X-100 lysis buffer (0,5% Triton X-100, 40 mM HEPES pH 7,6, 150 mM NaCl, 10% glycerol and freshly added EDTA-free protease inhibitor) for 10 min on ice. The lysates were centrifuged at 16800 g at 4 °C for 15 min and the supernatant was collected. After protein concentration determination as previously described, 2 mg of protein were incubated with 20 μL of M2 FLAG® sepharose beads at 4 °C and rotated in a Intelli mixer for 2 h. After pelleting the beads at 1000 g for 5 min at 4 °C, serial washes with decreasing concentrations of detergent (namely 0,5% 0,25%, 0,1%, and 0,0% Triton X-100) were performed. Protein elution was performed with 50 μL of 2x Laemmli buffer at 200 g at 45°C for 25 min. Alternatively, for subsequent DUB treatment or proteomics analysis, when indicated, MFN1$^{FLAG}$ or MFN2$^{FLAG}$ were eluted from FLAG beads by a 3x-FLAG peptide. For DUB treatment, 150 ng/μL 3x-FLAG peptide was dissolved in lysis buffer without Triton X-100 and with protease inhibitor and added to the FLAG beads for 30 min at 4 °C. For proteomics analysis, 150 ng/μL 3x-FLAG peptide was dissolved in 6 M urea/2 M thiourea and added to the FLAG beads for 30 min at 4 °C and the samples were analyzed by the CECAD Proteomics Facility.

## Immunoprecipitation of endogenous MFN2

HEK WT cells, untreated or treated for 4 h with 0.5 μM MLN-7243 were harvested with ice-cold PBS as previously described. Cells were resuspended and lysed with lysis buffer (1% NP-40, 50 mM tris pH 7.5, 150 mM NaCl, 2 mM EDTA, 5 mM NEM, 5% glycerol and freshly added PMSF to the final concentration of 1 mM). Cell lysis was performed with probe sonication (Microtip 2,5, 30% Amp, 2X10s) followed by 10 min of shaking at 4 °C. The lysates were centrifuged at 16800 g at 4 °C for 15 min and the supernatant was collected. After protein concentration determination as previously described, 7 mg of protein were incubated with an MFN2-specific antibody (rabbit from Cell Signaling, Supplementary Table 3) and the affinity resin with protein G immobilized (Protein G Sepharose 4 Fast Flow; GE Healthcare). After 3 h rotating at 4 °C, beads were washed three times in lysis buffer. Protein elution was performed with Laemmli buffer for

20 min shaking at 40 °C. 3% of the input and 100% of the eluate were analysed by SDS-PAGE.

## Deubiquitylase assay of whole cell lysates

HEK WT cells were harvested with ice-cold PBS as previously. Cells were resuspended and lysed with lysis buffer (1% NP-40, 50 mM tris pH 7.5, 150 mM NaCl, 2 mM EDTA, 5 mM NEM, 5% glycerol, and freshly added PMSF to the final concentration of 1 mM). Cell lysis was performed with probe sonication (Microtip 2,5, 30% Amp, 2X10s) followed by 10 min of shaking at 4°C. The lysates were centrifuged at 16800 g at 4°C for 15 min and the supernatant was collected. After protein concentration determination as previously described, 150 μg of protein were incubated for 10 min on ice with DTT to a final concentration of 10 mM. In parallel, purified USP21 (kindly gifted by Thomas Hermanns and prepared as in ref. 97) was diluted at 1:1 (final concentration 32uM) in DUB dilution buffer (25 mM tris pH 7.5, 150 mM NaCl, 10 mM DTT) and incubated at RT for 15 min. Cellular protein extracts were either left untreated or treated for 30 min at 37 °C with 5 μM of USP21 in DUB reaction buffer (50 mM tris pH7.5, 50 mM NaCl, 5 mM DTT). The reaction was stopped with 4X Laemlli buffer.

## Deubiquitylase assay of immunoprecipitated MFN1[FLAG] or MFN2[FLAG]

5 μM recombinant purified USP21 deubiquitylase (kindly gifted by Thomas Hermanns and prepared as in ref. 97) was incubated in DUB dilution buffer (25 mM tris pH 7.5, 150 mM NaCl, 10 mM DTT) for 15 min at RT and added to half of the eluate fraction obtained from previous MFN1[FLAG] or MFN2[FLAG] immunoprecipitation. Deubiquitylation was performed at 37 °C for 30 min and the reaction was terminated by adding 10 μL of 6x Laemmli buffer.

## Label-free quantification of whole cells proteome and interactome

For whole cell proteomics analysis of HEK WT, 1KO, 2KO and 2 + 2 cells, a total of 20 samples were prepared, 4 biological replicates of each of the five cell lines. The cells were lysed in urea buffer (5 M urea in 50 mM triethylammonium bicarbonate and freshly added protease inhibitor) and afterwards centrifuged for 15 min at 20000 g to remove cell debris and protein concentration was determined with the Bradford assay as previously described. 50 μg of protein were then prepared for proteomics analysis by an in-solution digestion. For this, DTT was added to a final concentration of 5 mM for 1 h at 25°C, followed by addition of chloroacetamide to a final concentration of 40 mM and incubation in the dark for 30 min. The protease LysC was added to an enzyme:substrate ratio of 1:75 and incubated at 25°C for 4 h, followed by overnight incubation of trypsin to an enzyme:substrate ratio of 1:75. To stop the digestion, formic acid was added and the eluate was loaded onto stage tips, for analysis by the CECAD Proteomics Facility.

For proteomics analysis of MFN1[FLAG] and MFN2[FLAG] interactome, a total of 4 samples including 1KO and 2KO cells as controls and 1KO + MFN1-3xFLAG and 2KO + MFN2-3xFLAG used for FLAG immunoprecipitation (n = 1 biological replicates) were used. 1KO cells stably expressing MFN1[FLAG] were treated with 10 μg/μl of tetracycline for 18 h for MFN1 expression to endogenous levels, while 2KO cells stably expressing MFN2[FLAG] did not require tetracycline induction for sufficient MFN2 expression to endogenous levels, due to leakiness of the system. FLAG immunoprecipitation was performed as described above, using 20 μL of M2 FLAG® sepharose beads. Then, for the elution step, 150 ng/μL 3x-FLAG peptide was dissolved in 6 M urea/2 M thiourea and added to the magnetic FLAG beads for 30 min at 4°C. Finally, the peptides were prepared from the eluate, by in-solution digestion, as described above, for analysis by the CECAD Proteomics Facility.

## LC-MS/MS analysis of whole cell samples

Samples were analyzed by the CECAD Proteomics Facility (University of Cologne) on Orbitrap Exploris 480 mass spectrometers (Thermo Scientific) coupled to Vanquish neo nano HPLC systems in trap-and-elute setup (Thermo Scientific). Samples were loaded onto a pre-column (Acclaim 5 μm PepMap 300 μ Cartridge) with a flow of 60 μl/min before being reverse-flushed onto an in-house packed analytical column (30 cm length, 75 μm inner diameter, filled with 2.7 μm Poroshell EC120 C18, Agilent). Peptides were chromatographically separated with an initial flow rate of 400 nL/min and the following gradient: initial 2% B (0.1% formic acid in 80 % acetonitrile), up to 6 % in 4 min. Then, the flow was reduced to 300 nl/min and B increased to 20% B in 50 min, up to 35% B within 27 min and up to 95% solvent B within 1 min while again increasing the flow to 400 nl/min, followed by column wash with 95% solvent B and re-equilibration to initial condition.

To generate a gas-phase fractionated library (Searle 2018), a pool of all samples was analyzed in six individual runs covering the range from 400 m/z to 1000 m/z in 100 m/z increments. For each run, MS1 was acquired at 60k resolution with a maximum injection time of 98 ms and an AGC target of 100%. MS2 spectra were acquired at 30k resolution with a maximum injection time of 60 ms. Spectra were acquired in staggered 4 m/z windows, resulting in nominal 2 m/z windows after deconvolution using ProteoWizard (Chambers 2012). For the samples, MS1 scans were acquired from 399 m/z to 1001 m/z at 15k resolution. Maximum injection time was set to 22 ms and the AGC target to 100%. MS2 scans ranged from 400 m/z to 1000 m/z and were acquired at 15 k resolution with a maximum injection time of 22 ms and an AGC target of 100%. DIA scans covering the precursor range from 400 - 1000 m/z and were acquired in 60 × 10 m/z windows with an overlap of 1 m/z. All scans were stored as centroid.

## Data analysis of proteomic samples

The gas-phase fractionated library was build using DIA-NN 1.8.1 (Demichev 2020) using a Swissprot human canonical fasta file (UP5640, downloaded 04/01/23) with settings matching acquisition parameters. Samples were analyzed in DIA-NN 1.8.1 as well using the previously generated library and identical fasta file. DIA-NN was run with the additional command line prompts "−report-lib-info" and "−relaxed-prot-inf". Further output settings were: filtered at 0.01 FDR, N-terminal methionine excision enabled, maximum number of missed cleavages set to 1, min peptide length set to 7, max peptide length set to 30, min precursor m/z set to 400, max precursor m/z set to 1000, cysteine carbamidomethylation enabled as a fixed modification. Afterwards, DIA-NN output was further filtered on >= 4 fragment ions per precursor, library q-value, global q-value <= 0.01 and unique peptides using the dplyr package in R. Finally, LFQ values were calculated using the DIA-NN R-package. Statistical analysis of results was performed in Perseus 1.6.15[98].

An unpaired two-sided t-test or one-way analysis of variance (ANOVA) were applied to identify significantly different proteins. The false discovery rate (FDR) was controlled to 0.05 (5%) using a permutation-based approach (number of permutations = 500, s0 = 0.1) in the Instant Clue software v.0.10.10.20210316[99].

All heat maps show z-scores of one-way ANOVA significantly changed peptides, following Euclidean clustering. Uniprot Gene Ontology annotations as well as MitoCarta 3.0 pathways[100] were added to the data and used for protein selection. For normalization of the mitochondrial mass, the protein groups positive for Mito-Carta 3.0 were selected and the median across all samples was adjusted to the global median (median of all samples). Then a two-sided t-test was calculated on the normalized data. The 1D enrichment[101] was performed in the Instant Clue software v.0.10.10.20210316[99].

## Proximity-ligation assay

The proximity-ligation assay was performed with the Duolink® In Situ Red Starter Kit Mouse/Rabbit (Sigma Aldrich). HEK cells were seeded on poly-L-lysine-coated coverslips and fixed in 4 % PFA in PBS for 10 min at RT, then washed three times with PBS. The cells were permeabilized by incubation with 0.01 % saponin in 3 % BSA for 30 min on a shaker at RT, then washed three times with PBS. Blocking was performed in a humidified chamber for 1 h at 37 °C. 40 μl of the desired primary antibodies (diluted 1:500 in antibody diluent) were added on the coverslips, ensuring that the entire surface was covered with the solution. The coverslips were incubated overnight at 4 °C, followed by washing eight times with approx. 200 μl of Wash Buffer A (prepared according to the manufacturer's specifications). 40 μl of the plus and minus probes (diluted 1:5 in antibody diluent) were added, incubated at 37 °C for 1 h, then washed eight times with buffer A. Next, 40 μl of the ligase (diluted 1:40 in 1x Ligation Buffer) were added and incubated at 37 °C for 30 min, followed by six washes with Buffer A. For the final amplification step, 40 μl of the polymerase (diluted 1:80 in 1x Amplification Buffer) were added and incubated at 37 °C for 100 min protected from light. After 10 washes with Wash Buffer B, the coverslips were mounted on microscope slides with Duolink® In Situ Mounting Medium with DAPI, dried for 15 min at RT, protected from light, and fixed to the slides with transparent nail polish. Imaging was performed on a Stellaris 5 LIAchroic inverse confocal microscope (Leica Microsystems).

Quantification of the proximity-ligation assay was performed using ImageJ v.1.53c. 50 cells per condition were manually outlined using the freehand drawing tool, based on the bright-field images, and saved as regions of interest (ROIs). The dots representing the protein interactions inside the ROIs were then counted either by eye (in the case of MFN2 + 20S, MFN2 + BAG2, MARCH5 + 20S) or by using the "Analyze particle" function in Fiji (MFN2 + HSPA8, MFN2 + PSMD11), preceded by a rolling ball background subtraction step and applying the "RenyiEnthropy" threshold.

## Oxygen consumption rate measurement

Cellular respiration was measured using the Agilent SEAHORSE XF incubator. For this, HEK cells were plated on 96-well plates and the following day the cells were washed three times with warm Assay Media (DMEM base media supplemented with 25 mM glucose, 1 mM pyruvate, 100 μM non-essential amino acids and 2 mM glutamine, pH 7.4) and incubated in the media for 1 h at 37 °C. After loading three injection ports of the sensor plate independently with oligomycin (1 μM), CCCP (0,3 μM) and a combination of Rotenone (0,5 μM) and antimycin A (0,5 μM), the oxygen consumption rate was measured 13 times over the course of 80 min. The OCR (pmol/min) was normalized to the protein concentration, determined as previously described.

## Crude mitochondria isolation

Crude mitochondria were isolated from HEK cells by incubation with M-Buffer (220 mM mannitol, 70 mM sucrose, 10 mM HEPES pH7.4, 1 mM EDTA, 1 mM PMSF, EDTA free protease inhibitor and 0,2% BSA) followed by swelling on ice for 15 min. Cell were homogenized at 100 g, 15 strokes, by a rotating Dounce Teflon homogenizer. This was followed by centrifugation for 5 min, at 500 g, at 4 °C. The supernatant was collected and centrifuged for 10 min, at 800 g, at 4 °C. The crude mitochondrial pellet was washed twice with M-Buffer (without BSA) and diluted in approximately 200 μl.

## Mitochondria isolation for blue native electrophoresis

Cells were harvested, washed twice with cold PBS, and resuspended in an ice-cold mitochondria isolation buffer (20 mM HEPES pH 7.6, 220 mM mannitol, 70 mM sucrose, 1 mM EDTA, supplemented with 0.2% fatty acid-free BSA). Following 20 min of incubation on ice, cells were homogenized using the rotational engine homogenizer (Potter S,

Sartorius; 30 strokes, 100 g). Heavy fractions were removed by centrifugation at 850 g, 10 min, 4 °C. Mitochondria were pelleted at 8500 g, 10 min, 4 °C, and washed with BSA-free mitochondria isolation buffer.

## Blue native electrophoresis

20 μg of the mitochondrial fraction were lysed with digitonin (6.6 g/g protein) for 15 min on ice with occasional vortexing and cleared from insoluble material for 20 min at 20,000 x g, 4 °C. Extracts were combined with Comassie G-250 (0.25% final), and mitochondrial respiratory supercomplexes were separated with Blue Native-PAGE using the 4–16% NativePAGE Novex Bis-Tris Mini Gels (Invitrogen) in a bis-tris/Tricine buffer system with cathode buffer initially supplemented with 0.02% G-250 followed by the 0.002% G-250. Separated mitochondrial complexes were transferred onto a polyvinylidene fluoride membrane (PVDF). Membranes were immunodecorated with antibodies, and mitochondrial respiratory supercomplexes were visualized using ECL-based signal detection.

## Cell growth measurement

Cellular growth was measured by cell confluency imaging using the live cell analyser JuLi brightfield system. One day after platting, cell confluency of HEK WT, 1KO and 2KO cells was measured over the course of 4 days (aprox. 100 h; 1 image/10 min).

## Protein aggregation isolation

For isolation of protein aggregates, cells from an 80-90% confluent 10 cm dish were collected with ice-cold phosphate-buffered saline (PBS) and centrifuged at 1000 g at 4°C for 5 min and washed once again with ice-cold PBS. The pellet was well resuspended in 1 mL Lysis Buffer (100 mM NaCl, 50 mM HEPES pH 7.4, 0.5% NP-40 and freshly added 1 mM PMSF, Benzonase and EDTA free protease inhibitor), sonicated for 10 sec (30% pulse) and centrifuged at 800 g at 4°C for 5 min and the supernatant was collected. Protein quantification was performed by using Bradford protein assay (Bio-Rad) according to the manufacturer's protocol and 150 μg of the whole cells were kept for further sample preparation as described above for Western blotting. The remaining protein lysate was used for fractionation of soluble/insoluble proteins and final volume of samples was made equal. Protein lysates were centrifuged at 21,000 g at 4°C for 30 min and the supernatant (soluble fraction) was collected. The pellet contained the insoluble fraction. The protein concentration of the soluble fraction was quantified by using Bradford protein assay (Bio-Rad) according to the manufacturer's protocol and 150 μg were used kept for further sample preparation as described above for Western blotting. The insoluble fraction (pellet) was further washed twice with 8 M urea wash buffer (8 M urea, 100 mM NaCl and 100 mM HEPES pH 7.4) and centrifuged at 21,000 g at 4°C for 30 min. Finally, the pellet was resuspended in 6x Laemmli Buffer (1 M tris-HCL pH 6.8, 60% glycerol, 4% SDS, 0,1% bromophenol blue and freshly added dithiothreitol (DTT)) and denatured at 200 g at 45°C for 20 min, for Western blotting.

## RNA isolation and quantitative PCR

Total RNA was extracted from cells using TRIzol reagent (Thermo Scientific, Dreieich, Germany), according to the manufacturer's instructions. For real-time PCR, 500 ng RNA was reverse transcribed into cDNA using the SuperScript™ III Reverse Transcriptase and oligo(dT) primers from Thermo Scientific. GoTaq qPCR Master Mix (Promega, Walldorf, Germany) was used to perform real-time quantitative PCR as recommended by the supplier and with the primer sets, listed in Supplementary Table 4. All assays were performed in triplicates on a CFX96 or CFX384 Thermo Cycler (Bio-Rad, Hercules, Feldkirchen, Germany). Transcript levels were quantified via the $2^{-\Delta\Delta Ct}$ method and normalized to HPRT mRNA transcript values.

## Mitochondrial membrane potential measurement

Mitochondrial membrane potential was assessed in HEK cells by their live staining with TMRE and Mitotracker Deep Red. First, HEK cells were stained with Mitotracker Deep Red (100 nM) for 15 min at 37 °C. Mitotracker Deep Red was washed twice with phenol-free DMEM containing all required supplements as detailed above. Next, cells were stained with TMRE (20 nM) for 30 min at 37 °C. After this incubation time, the cells were live imaged in a confocal microscope. For the positive control with CCCP treatment cells were treated with CCCP (20 μM) and immediately live imaged.

## MtKeima measurements

For lentiviral particles production, HEK293T were co-transfected with LV-Mito-Keima (MtKeima, kindly provided by Toren Finkel, Center for Molecular Medicine, National Heart, Lung, and Blood Institute, NIH, Bethesda, MD, USA) and the packaging plasmids pMDLg/pRRE, pRSV-Rev, pMD2.G (Supplementary Table 1) using the transfection reagent PEI. 8 h after transfection, the medium was replaced with fresh culture medium. Simultaneously, 2 million of the desired HEK293 cells lines were seeded. 48 h after transfection, the medium from the transfected HEK 293 T cells containing viral particles was collected, mixed with 6 μg/mL of Polybrene (Sigma-Aldrich), and delivered to the seeded HEK293 cells. After 36 h, the medium was changed. Each HEK293 cell line infected with pLVX-puro-MtKeima was seeded and after two days treated with 100 μM IU1 or DMSO (Sigma-Aldrich) for 24 h. The cells were collected by centrifugation at 200 g for 5 min, the pellet was washed twice with PBS and resuspended in 200 μL HBSS (Thermo Fisher Scientific) and 10 mM HEPES (Sigma-Aldrich) pH 7.4. MtKeima expressing cells were analyzed by flow cytometry (BD FACSAria™) equipped with a 405 nm and 561 nm laser.

## Statistics and reproducibility

No statistical method was used to predetermine the sample size. No data were excluded from the analyses. The experiments were not randomized. The investigators were not blinded to allocation during experiments and outcome assessment. All quantifications are presented as means ± standard deviation (SD) of different biological replicates (n indicating the number of biological repeats). Quantification of western blots was performed using ImageJ v.1.53c. Mass-spectrometry analysis was performed using Instant Clue v.0.10.10.20210316. Statistical significance was determined using ANOVA or t-students test calculations using GraphPad Prism v.9.4.1. The statistical test applied is specified in the respective figure and statistical analysis is displayed as non-significant (ns) with $p > 0.05$, * with $p < 0.05$, ** with $p < 0.01$, *** with $p < 0.001$ and **** with $p < 0.0001$. Exact $p$-values are presented within the figure legends.

Figure 1a outer mitochondrial staining with TOM20 and inner mitochondrial staining with ATP5α were performed independently more than 10 times with similar results. The experimental configuration presented was performed once.

Figure 3c was performed twice independently with similar results.
Figure 3d was perfored twice indepenedently with similar results.
Figure 3e was performed once.
Figure 7a-d was performed thrice independently with similar results.

## Reporting summary

Further information on research design is available in the Nature Portfolio Reporting Summary linked to this article.

## Data availability

All relevant data are included in the paper and/or the Supplementary Materials. The whole proteome data generated in this study has been deposited in the PRIDE database under accession code PXD058425. Source data are provided with this paper.

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

## Acknowledgements

We would like to thank J. Cassereau for the human primary skin fibro-blasts, T. Hermanns for providing us with purified USP21, T. Finkel for the pLVX-puro-Mito-keima plasmid, and M. Jevtic and M. Bergami for shar-ing of AAV viral vectors. We thank Angela Andersen for valuable com-ments on the manuscript's point by point. We are thankful to J.-W. Lackmann and the CECAD Proteomics Facility and to A. Schauss and the CECAD Imaging Facility for their excellent assistance, in particular B. Martiny and C. Jüngst, for the acquisition of the electron microscopy and confocal pictures, respectively. This work was supported by the Deutsche Forschungsgemeinschaft (DFG, German Research Founda-tion): in the frame of the SPP2453 – project number 541758846 to M.E.-H. and project number 541210481 to K.F.W.; in the frame of the CRC 1218 – TP A03, A11, and B01 to M.E.-H., E.Mo., and A.T., respectively, and Qua-lification fellowship to M.J.; in the frame of the German Excellence Initiative and Strategy – ZUK 81/1, to M.E.-H.; EXC 2030 – 390661388 – project number 269925409 – CECAD, to M.E.-H. and A.T.; and EXC 2033 – 390677874 – RESOLV, to K.F.W.; in the frame of large instrument grants – INST 216/1163-1 FUGG for the Mass-spectrometry at CECAD and INST 213/840-1 FUGG for the SR-SIM microscope to K.F.W., the last also

funded by the State Government of North Rhine-Westphalia. The work was also supported by a grant from the Center for Molecular Medicine Cologne (CMMC) – CAP14, to M.E.-H. and RPA02 to M.E.-H. and M. O.; from the Fritz-Thyssen foundation – 10.15.1.018MN, to M.E.-H.; from the Michael J. Fox Foundation – Grant ID: 021968 to K.F.W.; and the Bayer foundation – Otto-Bayer fellowship, to M.J.; from the Boehringer Ingelheim foundation – Plus 3 program, to M.E.-H.

## Author contributions

M.-B.B. and T.S. contributed equally to this work. M.J., S.A., M.-B.B., T.S., H.N., C.F., S.P., K.S. and E.Ma. performed the experiments. V.B. quantified protein aggregation. M.K., E.Z., A.T., A.C., K.F.W., E.Mo., M.O. and M.E.-H. provided experimental supervision and critical intellectual input. M.E.-H., M.J., and S.A. designed the study. K.H. provided critical intellectual input. M.J. and M.E.-H. wrote the manuscript, with input from all authors. E.Mo. and M.O. critically revised the manuscript. M.E.-H. coordinated the study.

## Funding

## Competing interests

The authors declare no competing interests.
