## [Transparent Peer Review file · Nature Communications]

Mitofusin 2 displays fusion-independent roles in proteostasis surveillance

Corresponding Author: Dr Mafalda Escobar-Henriques

Version 0:

Reviewer comments:

Reviewer #1

(Remarks to the Author)

In this manuscript the authors have analyzed potential novel function of the mitochondrial fusion protein MFN2. They document that MFN2 modulates proteome composition, and it does so through binding to chaperones, and to subunits of proteasome, thus preventing protein aggregation. The authors also document that MFN2 controls the stability of mitochondrial proteins MFN1, PINK1, TOM components. The manuscript is potentially interesting since it offers a potential novel molecular function of the mitochondrial protein MFN2. The major criticisms at this stage are the following: a) the authors should rule out that the identified partners of MFN2 (chaperones/proteasome subunits) are not an artifact, b) there is not mechanistic information on how MFN2 controls proteasome activity and how the effects on different mitochondrial proteins are opposite, and c) the study is limited to cultured cells, and no evidence is provided on whether this also occurs under in vivo conditions.

Major comments.

1. The authors show in Figure 2B that MFN2 KO cells show a differential proteome compared to WT or MFN1 KO cells. This does not tell whether there are differences in quality control unless this is further developed. In this connection, it would be key to show that the differences disappear in MFN2 KO cells upon expression of MFN2 (for instance MFN2FLAG protein. In addition, it would be interesting to show that MFN2 ablation mainly alters mitochondrial proteins or whether the alterations in the proteome are broader in nature. Let me also add that the global proteome should be presented in a supplementary table.
2. The identification of MFN2 and MFN1 partners was performed upon expression for 18 h of MFN2FLAG or MFN1FLAG in MFN2KO or MFN1 KO cells (triggered by tetracycline). Because tetracycline is known to cause mitochondrial proteotoxic stress, one could think that some of the effects detected in Figure 2C are a consequence of tetracycline rather than on MFN2FLAG. In this connection, the authors detected also proteasome partners upon MFN1FLAG immunoprecipitation (BAG2, STIP1), which suggests that some of the partners may be dependent on tetracycline and not just specific for MFN2FLAG. Based on this, the authors should provide convincing evidence that the chaperone/proteasome partners are indeed specific of MFN2 and not of the conditions used in the experimental assays. In addition, data in supplementary table do not show values of enrichment so it is hard to take any conclusions from it.
3. Another aspect that the manuscript misses is to validate some of the most exciting partners of MFN2, which should be done under endogenous conditions. Moreover, the impact of MFN2 ablation or its re-expression (MFN2FLAG) on proteasome activity should be also analyzed, and if so what is the nature of the mechanisms involved. If MFN2 binds to chaperones/proteasome subunits, it would be feasible to visualize the existence of proximity between those proteins in PLA assays.
4. The authors document and increased abundance of aggresomes in MFN2 KO cells. Do aggresomes contain ubiquitin, vimentin or gamma-tubulin? Is the increased aggresome abundance due to increased formation or instead is that due to reduced degradation? Are there changes in aggrephagy?
5. Western blot from Figure 4c does not clearly substantiate the quantification of MFN1. Please show a dedicated blot for MFN1.
6. The authors should provide some evidence to explain why the absence of MFN2 stabilizes mitochondrial proteins such as MFN1 or PINK1 and in contrast, it enhances the degradation of TOM20. There is not mechanistic information on how MFN2 affects mitochondrial stability in an opposite manner.
7. The manuscript message would be stronger if the authors could provide information on whether the process by which MFN2 impacts on proteasome components and proteolysis of mitochondrial proteins also occurs under in vivo conditions.
8. The study lacks statistical analysis.

Minor comments.

1. Data shown in Figure 1 are already known in the field so they may be omitted. In addition, they work in this figure in HeLa

cells that they were no longer used in the rest of the sections of the study.

Reviewer #2

(Remarks to the Author)

The manuscript by Joaquim et al. describes the role of MFN2, a known OMM protein of the mitochondrial fusion machinery, in cellular proteostasis. The authors show, that MFN2 is endogenously ubiquitinated even in the absence of stress and that it interacts with several chaperones and the proteasome machinery. They further explore a role of MFN2 in the activation of PINK1 and mitophagy, its role in protecting MFN1 degradation, problems in the import machinery, and the role of proteostasis in NAFDL and CMT2A.

Although I found a lot of interesting information presented in this manuscript, at this stage the manuscript is a bit disjointed. The manuscript would benefit from streamlining and focusing on one aspect of MFN2 function (apart from mitochondrial fusion). No statistical analysis of the presented data was performed, please correct.

Major comments/questions:

The authors use both HeLa and HEK knock-out cells, however, jump from one to another, without clearly stating if there are any differences in 1KO, 2KO or DKO in these two cell lines. Re-organization of the figures (one cell line main figures, second cell line in Supplemental) and description of similarities/differences would be highly appreciated. Plus, some key information is missing: WB for KO in HeLa; cristae in HEK seem to be better preserved in 1KO and 2KO than in HeLa; so is the morphology of 1KO as quantified in S1D); Fig. S2C,D: the images shown are HEK, however quantification presented is from HeLa cells. Also, data presented in Fig1 and most of the Supplemental Figs.1-5 are well established using MEFs. I understand that it is important to show this effect in human cell lines, but could be all simplified.

Authors performed quantitative MS analysis of 2KO vs. WT cells, however they did not explore this important source of information (aside from effects on OXPHOS proteins). They did not even discuss how many proteins were affected (either increased or decreased). It would be interesting to know, if the proteomic analysis confirms the accumulation of "aggregated/non-degraded" proteins? What other pathways (aside from OXPHOS) were affected/decreased in 2KO? In Fig. 5 the authors look at several mitochondrial proteins in the 2KO cells by Western-blot analysis, why were these particular proteins selected? What happens with MFN1 in 2KO cells, a Western-blot similar to those in Fig. 5 would be informative? The authors looked at degradation rate of MFN1 in 2KO and WT cells upon CHX treatment, however did not show if proteasomal inhibition changes the levels of MFN1, as it does for MFN2? OPA1 plays a role both in mitochondrial fusion and fission, dependent on the ratio of the short and long forms of the protein. Was there a shift in the ratio of the short/long forms of OPA1 in 2KO cells? The authors claim that PINK1 accumulated at the OMM, however show, that epoxomicin inhibition of the proteasome leads to the cleavage of PINK1 by PARL (although to a lower level than in WT cells). What are the levels of PARL in mitochondria in 2KO cells, is the effect on PINK1 cleavage due to the low levels of PARL? Also, the WB analysis in Fig.5 was done on whole-cell extracts, normalized to Ponceau staining. However, if the size/mass of mitochondria was changed in 2KO, this analysis would be invalid. A WB of mitochondrial extracts should be presented.

It is obvious that the effect of MFN2 ablation on mitochondrial biology is very complex. As mitochondrial import machinery is affected in 2KO cells (Fig. 5), the decrease in other mitochondrial proteins might be due to the decrease in the import, followed by proteasomal degradation of unimported proteins and not really linked to MFN2 directly. Lack of MFN2, as the link to proteasome, should rather lead to accumulation of selective proteins (as the authors show for some proteins) and not to an increased degradation of proteins? Could the authors comment/discuss?

The authors show a co-immunoprecipitation of MFN2 with the proteasome. Is the interaction of MFN2 with proteasome lost upon treatment with USP21? Is MFN2 forming a "link" between mitochondria and proteasome or is the interaction due to the fact that MFN2 is degraded by the proteasome? A very recent paper by this group shows that MFN2 (and especially the CMT2A mutants) is ubiquitinated by UBE4B, and targeted to proteasome, did the authors look at the levels E1/E2/E3/E4 ubiquitin ligases (including UBE4B) in 2KO cells? Are proteasomal subunits and chaperones affected?

During apoptosis, the authors claim, that MFN1 and PINK1 degradation are prevented in 2KO, while MCL1 is not affected. However, no quantification is provided for PINK1 and MCL1? If MFN2 provides a platform for a specific degradation of proteins by proteasome it would be interesting to analyze the changes in the whole cell proteome under ActD treatment of 2KO vs. WT cells, although I understand this might be beyond the scope of this revision. However, the authors could look at several proteins of OMM (FIS1, MIEF1, DRP1) known to be linked to apoptosis and proteasome.

Is it possible to use the word "dominance" when describing the fact that MFN2 expression rescues 1KO and DKO? This sentence in the discussion: "These observations underline a dominant role of MFN2 over MFN1 in determining mitochondrial structure in human cells, being MFN1 probably more prominent than MFN2 in hyperfusion events (51–55).", is not very clear.

The effect of siDRP1 on the resolution of protein aggregates is very intriguing, however this argues against the role of MFN2 in the aggregation? Authors should try to promote mitochondrial elongation/total fragmentation in different ways (CHX treatment, starvation, CCCP) to see if MFN2 or mitochondrial morphology play a role in aggregation.

MFN2 has been shown to play a role in mitochondria-ER contacts, but there is almost no mention or analysis done in this respect.

Minor comments:

Authors should not use red/green combination for images.

Fig. 2b – labels in the figure state WT, MFN1, MFN2 but should be WT, 1KO, 2KO

Fig. S6A – The size difference between tagged and untagged MFN1/MFN2 seem to be much bigger than in Fig. S5A for MFN2. Did authors use a single FLAG tag, or a multiple FLAG? As MFN1 and tagged-MFN2 run at the same molecular weight in the figure, a blot with anti-FLAG antibody would be useful to confirm that indeed tagged-MFN2 is expressed.

Fig. 7a – a legend for the graph is not very clear, as it is not obvious which is WT, 1KO and 2KO, although presumably it lines up with the blot.

Line 131/132 Taken together, our results confirm the need of both mitofusins for mitochondrial tubulation. – “tubulation” is not the correct word, as they do not tubulate mitochondria, but play a role in fusion.

Line 225 rescued all OXPHOS defects of 2KO cells (Supplementary Fig. S6). - Should be Fig. S5

Line 258 proximity to the mitochondria or distributed throughout the cytoplasm (Fig. 3a). - Should be Fig. 3c

Line 313 4b, c and Supplementary Fig. 9a). - Should be Fig. 9c

Line 325 Interestingly, the mitochondrial inner membrane fusion factor OPA1 and the mitochondrial fission factor DRP1 were decreased and increased in 2KO cells, respectively, supporting the loss of mitochondrial fragmentation upon MFN2 ablation (Fig. 5a). – I presume the authors mean “supporting the mitochondrial fragmentation upon MFN2 ablation”.

Reviewer #3

(Remarks to the Author)

This paper presents an exciting story which is further substantiates the recent discoveries from the field of proteostasis and organellar quality control linked to the mitochondria. It adds another important player in this process, Mfn2, raising a possibility that this protein is a primarily involved in a process different than mitochondrial fusion. The recent years opened a very interesting and vibrant field of the links between cellular proteostasis and mitochondria, but neither introduction nor discussion reflect it although the current, really interesting and well done study is located placed in this area. Thus, I recommend to modify these parts for better appreciation of this work in the field.

The manuscript provides a comprehensive set of data, the experiments and findings are in general very convincing. The suggested experiments should improve this work:

- the most important effects characterizing the proteostasis changes in Mfn2KO (aggregates, a decrease in mitochondrial proteins, i.e. import and the PINK angle) should be performed in the complementation system, to avoid some cross-talk

- the nature of aggregates should be analyzed by bioinformatic means, with attention to perhaps small qualitative differences from the control with the proteasome inhibitors, especially that the mitochondrial import machinery is decreased in Mfn2KO

- in line with the point above, how the decrease of import machinery affects the aggregation and proteasome binding? how the import deficiency contributes to it?

- the proteasome is presented as it would be bound to the surface of mitochondria, based on its interaction with Mfn2. This however is not the case, the large body of the literature shows a lack or no enrichment on the mitochondrial surface under wild type conditions. The situation is changing (slightly) upon stress (the Sladowska et al., and recent Kim et al. 2023). Could authors comment how they explain the conundrum of seeing proteasome subunits with Mfn2-Flag, given it is properly located in the mitochondrial OM?

I am not certain that the name "sentinel" exactly reflects the function of this protein- this is a suggestion for authors consideration.

Version 1:

Reviewer comments:

Reviewer #1

(Remarks to the Author)

The revised version presented by the authors has responded the major questions raised by the Reviewer, and as a result, the manuscript has improved substantially. I only have some minor issues:

1. In Table S2 the authors include the protein interactome of both mitofusins. They also show enrichment data for the different proteins expressed as a log₂ ratio MFN1 or 2 and DMSO. They also show statistics, which is shown as –log t test p value. This is an unusual way to show the statistics and it may be easier to the reader to use p value and adjusted p value.
2. Analysis of the mitophagy activity shown in Figure 2F should include a group in the presence of bafilomycin A1. Otherwise, they should tone down their conclusions on the impact of MFN2 ablation on mitophagy. In addition, this is not the major observation of this study, and should be omitted from the abstract.
3. Line 402: the authors should state why they used MLN-7243. There is no mention on this, which does not facilitate the reading of the manuscript.

Reviewer #2

(Remarks to the Author)

The authors have now significantly improved the manuscript. They show that MFN2 plays a role in cellular proteostasis, via its interaction with cellular chaperones and with the proteasome. I do have some comments, that need to be addressed before this manuscript is suitable for publication:

- Authors claim, that Complex II is not affected in 2KO (line 167/168), however their proteomics analysis clearly shows a reduction in protein levels of all subunits of this complex (Supplementary Fig. 5a). Can authors comment?
- Fig. 2f and Supplementary Fig. 6 also shows K357N MFN2 variant, however there is no mention of this in the text?
- Lines 237-240 and Supplementary Fig. 10a: Why is there a staining with MFN2 antibody in 2KO cells that is nuclear? There should not be any PLA signal in 2KO cells, as they do not express MFN2, so this statement is misleading. To show the specificity of MFN2 interaction with proteasome, another OMM protein (such as MFN1) +PSMD11 PLA should be used as a negative control. Did authors perform MFN1-PSMD11 PLA in either WT or 2KO cells? Also, PSMD11 staining in 2KO cells is nuclear, while it is cytosolic in WT, any explanation why?
- In some figures, loading controls are used twice (Supplementary Fig. 5D-SDHA; Supplementary Fig. 7h- Ponceau). Authors should mention this in the legend, or remove one copy, as they might be criticized for image duplication.
- Some immunofluorescence images are not very crisp. Could authors improve the contrast? Supplementary F3b, Fig. 3b, f, Supplementary 10a

Reviewer #3

(Remarks to the Author)

I appreciate the current version of the manuscript and fully support it.

Version 2:

Reviewer comments:

Reviewer #1

(Remarks to the Author)

The authors have adequately answer my last questions so the manuscript reads well and it is a relevant piece of work.

Reviewer #2

(Remarks to the Author)

The authors have now satisfactorily answered all my questions/comments.i

Reviewer #1 (Remarks to the Author):

In this manuscript the authors have analyzed potential novel function of the mitochondrial fusion protein MFN2. They document that MFN2 modulates proteome composition, and it does so through binding to chaperones, and to subunits of proteasome, thus preventing protein aggregation. The authors also document that MFN2 controls the stability of mitochondrial proteins MFN1, PINK1, TOM components. The manuscript is potentially interesting since it offers a potential novel molecular function of the mitochondrial protein MFN2. The major criticisms at this stage are the following: a) the authors should rule out that the identified partners of MFN2 (chaperones/proteasome subunits) are not an artifact, b) there is not mechanistic information on how MFN2 controls proteasome activity and how the effects on different mitochondrial proteins are opposite, and c) the study is limited to cultured cells, and no evidence is provided on whether this also occurs under in vivo conditions.

We are grateful to the reviewer for recognizing our findings on a novel molecular function of MFN2 and for the many valid suggestions that have enabled us to improve our manuscript.

Major comments:

Point 1 (P1): The authors show in Figure 2B that MFN2 KO cells show a differential proteome compared to WT or MFN1 KO cells. This does not tell whether there are differences in quality control unless this is further developed. In this connection, it would be key to show that the differences disappear in MFN2 KO cells upon expression of MFN2 (for instance MFN2FLAG protein).

In addition, it would be interesting to show that MFN2 ablation mainly alters mitochondrial proteins or whether the alterations in the proteome are broader in nature. Let me also add that the global proteome should be presented in a supplementary table.

Response 1 (R1): We totally agree with this comment and now present mass spectrometry data of MFN2 KO cells upon stable re-expression of MFN2 (named as 2+2 cells), which rescues all phenotypes observed (new data in Fig. 1 and Supplementary Fig. 4). Moreover, these mass spectrometry results are now available in the Supplementary Table 1.

We thank reviewer #1 for the interesting question whether MFN2 KO mainly alters mitochondrial proteins. Indeed, most of the changes reflect a decrease in protein levels, particularly of mitochondrial proteins (new data Fig. 1c and Supplementary 4d) (consistent with reduced mitochondrial mass, Supplementary Fig. 4c), but over 30% of proteins go up, and these are enriched for non-mitochondrial proteins (Fig. 1c and Supplementary 4b). Briefly, we found that 1421 proteins were altered by MFN2 KO: 912/1421 (64%) proteins were downregulated (53% of these (482/912) are mitochondrial proteins), whereas 509/1421 (36%) proteins were upregulated (6% of these (29/509) are mitochondrial proteins) (Supplementary Fig. 4b).

Pathway enrichment analysis shows downregulation of OXPHOS subunits and mitochondrial ribosomal components and an upregulation of fatty acid oxidation and ROS/glutathione pathways (Fig. 1d,e and Supplementary Fig.4 c-f). This agrees with reports where MFN2 loss inhibits mitochondrial respiration¹⁻⁵ and upregulates fatty acids⁸. These findings also agree with reports where deficiencies in OXPHOS and complex V were shown to upregulate fatty acid & ROS^{6,7}.

Importantly, we did not observe an effect of MFN2 KO on the abundance of chaperones, proteasomal subunits, and other quality control components. Therefore, as we now further develop in the manuscript, MFN2 cooperates with chaperones and the proteasome (please see R4) without affecting their protein levels or localization (please see R3). This is consistent with the proposed role of MFN2 in constitutively surveilling the cellular status at the mitochondrial surface.

P2: The identification of MFN2 and MFN1 partners was performed upon expression for 18 h of MFN2FLAG or MFN1FLAG in MFN2KO or MFN1 KO cells (triggered by tetracycline). Because tetracycline is known to cause mitochondrial proteotoxic stress, one could think that some of the effects detected in Figure 2C are a consequence of tetracycline rather than on MFN2FLAG. In this connection, the authors detected also proteasome partners upon MFN1FLAG immunoprecipitation (BAG2, STIP1), which suggests that some of the partners may be dependent on tetracycline and not just specific for MFN2FLAG. Based on this, the authors should provide convincing evidence that the chaperone/proteasome partners are indeed specific of MFN2 and not of the conditions used in the experimental assays. In addition, data in supplementary table do not show values of enrichment so it is hard to take any conclusions from it.

R2: We thank the reviewer for this comment and appreciate the opportunity to clarify our approach.

MFN2^{FLAG} in MFN2KO cells stably expressing MFN2^{FLAG} is actually expressed at similar levels to endogenous MFN2 in wild type cells, even in the absence of tetracycline, because the CMV promoter is leaky in this cell line (Supplementary Figs. 4a and 9a). Thus, in this case no tetracycline was added. This therefore excludes the concern that MFN2^{FLAG} interactors are an artifact of tetracycline addition. Importantly, as nicely suggested by the reviewer (in P3), we now confirmed the specificity of the interactions with proximity-ligation data under endogenous conditions (please see R3).

In the case of MFN1KO cells expressing MFN1^{FLAG}, we used 10 µg/µL of tetracycline to induce MFN1^{FLAG} expression. However, MFN1^{FLAG} didn't bind the proteasome or HSC70/HSP90, reinforcing that in our assay tetracycline didn't cause unspecific protein binding to the proteasome. It is possible that BAG2 and STIP1 bind MFN1 indirectly through MFN2, given that MFN2 co-immunoprecipitated with MFN1^{FLAG}. Importantly, we did not detect proteostasis defects in MFN1 KO cells, so we didn't pursue any of the MFN1^{FLAG} binding partners.

The values of enrichment from this mass spectrometry analysis are now available in the Supplementary Table 2.

P3: Another aspect that the manuscript misses is to validate some of the most exciting partners of MFN2, which should be done under endogenous conditions. Moreover, the impact of MFN2 ablation or its re-expression (MFN2FLAG) on proteasome activity should be also analyzed, and if so what is the nature of the mechanisms involved. If MFN2 binds to chaperones/proteasome subunits, it would be feasible to visualize the existence of proximity between those proteins in PLA assays.

R3: We totally agree on the importance of demonstrating the interactions on an endogenous level and are very thankful for this suggestion. We established proximity-ligation assays in our cellular models, which confirmed the interaction of endogenous MFN2 with the 20S and 19S proteasome (new data Fig. 3b), and also with HSC70 (new data Fig. 5a), BAG2 (new data Supplementary Fig. 12c) and MARCH5 (new data Supplementary Fig. 13a).

To investigate the impact of MFN2 ablation on proteasome activity we compared global ubiquitylation levels in WT and in MFN2 KO cells, both by western blot (new data Supplementary Fig. 11e and western blot with P4D1 antibody after cellular fractionation in Response Fig. 2, after point R4) and by immunostaining (new data Supplementary Fig. 11f). We detected a mild increase of global protein ubiquitylation in MFN2KO cells, along with stabilization of MFN1 (Fig. 4a,b) and PINK1 (Fig. 4c). However, MFN2KO didn't affect the levels of the proteasome assembly proteins identified in other studies of mitochondrial dysfunction⁹ (new data Supplementary Fig.

10c). In addition, MFN2KO didn't affect the levels and localization of 20S proteasome, p97 or BAG2 (Response Fig.1, below). In conclusion, MFN2 seems to play a mild role in global protein turnover without globally affecting proteasomal activity, reinforcing the impact of mitochondrial regulation in cellular quality control.

Response Fig. 1: Confocal images after immunostaining with the 20S proteasome (upper panel, in green), p97/VCP (middle panel, in green) and BAG2 (lower panel, in green) and the mitochondrial protein TOM20 (in red, in all panels) of HEK WT and 2KO cells.

P4: The authors document and increased abundance of aggresomes in MFN2 KO cells. Do aggresomes contain ubiquitin, vimentin or gamma-tubulin?
Is the increased aggresome abundance due to increased formation or instead is that due to reduced degradation? Are there changes in aggrephagy?

R4: Co-immunostaining of protein aggregates (with Proteostat) and gamma-tubulin show that the MFN2KO aggregates do not substantially enrich this intermediate filament protein, which is consistent with its localization with the aggresome directly at the centromere (new data Supplementary Fig. 15a).

In contrast, and very interestingly, vimentin caged both the aggresome, present in proteasomal-blocked WT cells, and the smaller protein aggregates, present in proteasomal-blocked WT and in MFN2KO cells (new data Supplementary Fig. 15b).

For ubiquitin the same approach was extensively attempted. However, Proteostat is a dye with a very large excitation/emission wavelength spectrum. Hence, co-localization analysis, i.e. co-staining, is only possible with very strong primary antibodies, which is not the case for ubiquitin. Therefore, we cannot conclude if the protein aggregates contain ubiquitin. We also tried to answer this question via cellular fractionation. However, while ubiquitin was enriched in the insoluble proteins upon proteasomal blockage, we couldn't detect any ubiquitin signal in MFN2KO cells (Response Fig. 2, below). Therefore, as detailed also in our response R2 to reviewer #3, we aim at optimizing the aggregate extraction in future experiments, to be able to characterize the proteome of MFN2KO protein aggregates, including its ubiquitylation properties. Nevertheless, given that ubiquitin is required for aggregate formation (please refer to the next paragraph), they will likely also contain ubiquitin.

Response Fig. 2: Western blot analysis of whole cell lysates, soluble fractions and 8M Urea-resistant insoluble fractions from HEK WT cells, untreated (WT), treated with Epoxomycin (Epoxo) or transfected with the aggregation-prone protein polyQ97-GFP (Q97-GFP) and MFN2KO cells, immunoblotted with anti-ubiquitin (α P4D1) and anti-PINK1 antibodies. Staining of total protein with PoS was used as loading control.

In our study, protein aggregation was detected with the Proteostat dye. This non-biased approach clearly presents the big advantage of allowing us to detect aggregation propensity of cellular proteins, globally, endogenously and regardless of their nature.

Since the composition of MFN2KO protein aggregates is not known, pulse-chase experiments could not be used to determine if the increased aggresome abundance is due to increased formation or instead is that due to reduced degradation. Therefore, to address this very important question, we targeted cellular pathways that impact formation and clearance of protein aggregates.

First, HSC70 inhibition with VER-155008 increased aggregate formation in MFN2KO cells (new data Fig. 5f). Given that HSC70 assists the import of newly synthesized mitochondrial proteins, these results suggest that MFN2 together with HSC70 prevent aggregate formation. Second, cycloheximide treatment, which inhibits protein translation, attenuated protein aggregation in MFN2KO cells (new data Fig. 5g). This is consistent with increased formation of protein aggregates from newly synthesized proteins. Third, inhibition of ubiquitylation in MFN2KO cells, by inhibiting the E1 enzyme UBA1 with MLN-7243, reduced these aggregates (new data Fig. 6c). This suggests that ubiquitin is required for protein aggregates formation in MFN2KO cells.

Regarding the clearance of aggregates, we investigated the importance of autophagy and proteasomal turnover. Inhibition of autophagy with Bafilomycin A did not alter aggregation propensity in MFN2KO cells (new data Fig. 6a). We also investigated p62, an abundant autophagy marker that localizes to protein aggregates and signals their clearance by aggrephagy. However, p62 is not upregulated in MFN2KO cells (Response Fig. 3). We infer that the clearance of protein aggregates by autophagy is not impaired in MFN2KO cells. In contrast, inhibition of the proteasome with MG132 aggravated protein aggregation in MFN2KO cells, suggesting a cooperation between MFN2 and the proteasome (new data Fig. 6b). Moreover, similarly to MFN2KO cells, protein aggregation upon MG132 treatment is blocked by

cycloheximide¹⁰. These new results indicate that MFN2 affects both the formation and clearance of protein aggregates, consistent with its physical interaction with major components on both the folding and turnover branches of protein quality control.

Response Fig. 3: Upper: Confocal images after immunostaining with p62 (in green) and the mitochondrial protein TOM20 (in red) of HEK WT and MFN2KO cells, untreated or treated with Bafilomycin A (100nM, for 2h). Lower: Western blot analysis of total cell lysates from HEK WT and MFN2KO, untreated (-) or treated (+) with Bafilomycin A (100 nM, 4 h), immunoblotted with anti-LC3. Staining of total protein with PoS was used as loading control.

P5: Western blot from Figure 4c does not clearly substantiate the quantification of MFN1. Please show a dedicated blot for MFN1.

R5: As requested by the reviewer we now added a dedicated blot for MFN1 (Fig. 4a).

P6: The authors should provide some evidence to explain why the absence of MFN2 stabilizes mitochondrial proteins such as MFN1 or PINK1 and in contrast, it enhances the degradation of

TOM20. There is not mechanistic information on how MFN2 affects mitochondrial stability in an opposite manner.

R6: The apparent contradictory action of MFN2 on the overall levels of these mitochondrial proteins reflects its different roles.

On one hand, ubiquitylated MFN2 binds to the proteasome and promotes proteasomal turnover of its cognate MFN1 and of PINK1 at the OMM. MFN2 also controls global ubiquitylation levels to a certain degree, and its depletion upregulates 30% of proteins, from which most are non-mitochondrial (Supplementary Fig. 4d). This suggests that MFN2 might promote proteasomal turnover of a large subset of proteins besides MFN1 and PINK1. Future pulse-chase experiments will allow determining the identity and properties of these proteins, an analysis we believe is beyond the scope of this revision.

On the other hand, MFN2KO causes an overall decrease of mitochondrial mass, and reduces the levels of TOM20 and OXPHOS subunits (Fig. 1, Supplementary Fig. 4, 5, 6, Fig. 2, Supplementary Fig. 7). It is well-established that MFN2 depletion has a strong impact on OXPHOS, but the reasons behind this phenotype are still not known. Notably, the loss of mitochondrial mass and OXPHOS subunits is not due to an increased proteasomal turnover (Supplementary Fig. 6f, 7f). Given that it is also not caused by decreased transcription (Supplementary Fig. 6e, 7e), it might reflect a post-translational effect, e.g. impaired targeting to mitochondria, followed by aggregation. This question will be investigated with future experiments that are currently being optimized and we believe are beyond the scope of this revision. As pointed out by reviewer #2, MFN2 is a complex protein with many biological functions that are difficult to fully elucidate.

P7: The manuscript message would be stronger if the authors could provide information on whether the process by which MFN2 impacts on proteasome components and proteolysis of mitochondrial proteins also occurs under *in vivo* conditions.

R7: To understand if the phenotypes observed in MFN2KO cells were reproduced *in vivo*, we capitalized on published murine models of Mfn2 depletion^{3,11}. We first employed primary cultures of cortical neurons lacking Mfn2, which showed decreased levels of Tom20 by Western Blot analysis (Fig. 2b), in agreement with the results obtained in our human cellular model. Unfortunately, and despite having tested several PINK1 antibodies, we failed to detect any reliable signal for Pink1 even upon treatment of these cultures with proteasomal inhibitors or with CCCP. In addition, we were not able to investigate the presence of protein aggregates with the Proteostat dye in these cultures, due to overlap in the excitation spectra of Proteostat dye and the reporter GFP which was expressed in the AAV-Cre we employed to induce MFN2 ablation *in vitro*.

Additionally, we used a validated model of mitochondrial dysfunction lacking Mfn2 specifically in cerebellar Purkinje neurons (Mfn2cKO mice)^{3,11} to evaluate protein aggregation. In this mouse model, Purkinje neurons exhibit severe signs of mitochondrial dysfunction despite a preserved viability at 8 weeks of age³. Therefore, we stained cerebellar slices of 8-week-old control (CTRL) or Mfn2cKO mice with Proteostat. We have tested several staining conditions, including different concentrations of Proteostat, different permeabilization methods of the tissue sections, and Proteostat labeling before or after immunostaining for the Purkinje cell marker Calbindin () (Response Fig. 4). However, in all tested conditions we mainly observed unspecific labelling of both CTRL and Mfn2cKO slices by Proteostat staining, thus making usage of the dye unreliable for aggregates detection in these samples. We have thus decided not to include these data in the revised manuscript.

Finally, with the goal of putting our discoveries in the context of disease relevant models, we collected fibroblasts from Charcot-Marie Tooth Type 2A patients with different

MFN2 mutations. Interestingly, we detected *increased protein aggregation in CMT2A patient fibroblasts* (Fig. 7), substantiating the physiological relevance of our findings.

Response Fig. 4: Upper panel: Proteostat staining (red) of 8-week-old cerebellar slices of Mfn2cKO mice. Calbindin (grey) is used to label Purkinje neurons, the brain cell type lacking Mfn2 in this mouse model. Several technical approaches were performed for optimization of the Proteostat staining of brain sections, as described on the right side of the panel.

Lower panel: Calbindin (white) and Proteostat (red) co-staining of 8-week-old cerebellar slices of Purkinje neurons of control (CTRL) or Mfn2cKO mice.

P8: The study lacks statistical analysis.

R8: We apologize for this and have now included statistical analysis to all quantifications.

Minor comments.

Minor point 1(MP1): Data shown in Figure 1 are already known in the field so they may be omitted. In addition, they work in this figure in HeLa cells that they were no longer used in the rest of the sections of the study.

MR1: We have largely simplified the role of MFN1 and MFN2 in mitochondrial morphology. We also placed the morphology in HEK cells as Supplementary Fig. 1 and the morphology in HeLa cells as Supplementary Fig. 2. This allows to underline the conservation in the different roles of MFN1 and MFN2. It is correct that the HeLa cell model is not very much used. However, HeLa cells were required for the experiments presented in Fig.4b,c,d, related to induction of apoptosis.

Reviewer #2 (Remarks to the Author):

The manuscript by Joaquim et al. describes the role of MFN2, a known OMM protein of the mitochondrial fusion machinery, in cellular proteostasis. The authors show, that MFN2 is endogenously ubiquitinated even in the absence of stress and that it interacts with several chaperones and the proteasome machinery. They further explore a role of MFN2 in the activation of PINK1 and mitophagy, its role in protecting MFN1 degradation, problems in the import machinery, and the role of proteostasis in NAFDL and CMT2A.

Although I found a lot of interesting information presented in this manuscript, at this stage the manuscript is a bit disjointed. The manuscript would benefit from streamlining and focusing on one aspect of MFN2 function (apart from mitochondrial fusion). No statistical analysis of the presented data was performed, please correct.

We are grateful to the reviewer for the very many helpful suggestions that have enabled us to improve our manuscript and for the streamlining suggestion which we agree helps to highlight the manuscript novelties.

With the goal of streamlining the study, we have now largely simplified the basic characterization of the MFN1&2 KO in HEK and HeLa cell lines (as explained below in R1). Moreover, we kept the aspects related with lipotoxicity for a later study, as we feel this requires further investigation.

Major comments/questions:

Point 1 (P1): The authors use both HeLa and HEK knock-out cells, however, jump from one to another, without clearly stating if there are any differences in 1KO, 2KO or DKO in these two cell lines. Re-organization of the figures (one cell line main figures, second cell line in Supplementary) and description of similarities/differences would be highly appreciated.

Plus, some key information is missing: WB for KO in HeLa; cristae in HEK seem to be better preserved in 1KO and 2KO than in HeLa; so is the morphology of 1KO as quantified in S1D; Fig. S2C,D: the images shown are HEK, however quantification presented is from HeLa cells.

Also, data presented in Fig1 and most of the Supplementary Figs.1-5 are well established using MEFs. I understand that it is important to show this effect in human cell lines, but could be all simplified.

R1: We thank the reviewer for these comments. We now present the characterization of mitofusin KOs in different cell lines in separate figures: HEK cells in Supplementary Fig. 1 and HeLa cells in Supplementary Fig. 2. As requested, we added the WB for KO in HeLa cell (Supplementary Fig. 2a), and briefly discussed similarities/differences between both cellular models but kept this part of the manuscript very simple. Indeed, cristae in HEK are better preserved than in HeLa, a fact commonly acknowledged in the field, which might be due to differences between cancer and non-cancer cell lines. To further simplify the manuscript, we removed the supplementary information regarding the morphology of other cellular organelles in the MFN2KO cells because, as the referee has mentioned, it is well established in other models. However, we prefer to keep the analysis of OXPHOS complexes in the manuscript, since it is a major phenotype of MFN2KO cells (Fig. 1d,e).

P2: Authors performed quantitative MS analysis of 2KO vs. WT cells, however they did not explore this important source of information (aside from effects on OXPHOS proteins). They did not even discuss how many proteins were affected (either increased or decreased). It would be interesting to know, if the proteomic analysis confirms the accumulation of “aggregated/non-degraded” proteins? What other pathways (aside from OXPHOS) were affected/decreased in 2KO?

R2: We entirely agree with reviewer #2 and are thankful for this suggestion. We now further expand this analysis with new MS data of MFN2KO cells upon stable expression of MFN2^{FLAG}, in Fig. 1 and accompanying Supplementary Fig. 4, and discuss all aspects suggested by the reviewer. Moreover, mass spectrometry data is now available in the Supplementary Table 1.

We now report that most of the changes in MFN2 KO cells reflect a decrease in protein levels, particularly of mitochondrial proteins (new data Fig. 1c and Supplementary 4d) (consistent with reduced mitochondrial mass, Supplementary 4c), but over 30% of proteins go up, and these are enriched for non-mitochondrial proteins (Fig. 1c and Supplementary 4d). Briefly, we found that 1421 proteins were altered by MFN2 KO: 912/1421 (64%) proteins were downregulated (53% of these (482/912) are mitochondrial proteins), whereas 509/1421 (36%) proteins were upregulated (6% of these (29/509) are mitochondrial proteins) (Supplementary Fig. 4b).

Pathway enrichment analysis shows downregulation of OXPHOS subunits and mitochondrial ribosomal components and an upregulation of fatty acid oxidation and ROS/glutathione pathways (Fig. 1d,e and Supplementary Fig.4 c-f).

Aside from OXPHOS, pathway analysis shows the downregulation of mitochondrial ribosomal and protein import components. Moreover, we observe an upregulation of fatty acid oxidation and ROS/glutathione pathways. This agrees with reports where MFN2 loss inhibits mitochondrial respiration¹⁻⁵ and upregulates fatty acids⁸. These findings also agree with reports where deficiencies in OXPHOS and complex V upregulates fatty acid & ROS^{6,7}.

Mass-spectrometry analysis could unfortunately not be used to confirm the aggregation-prone proteins that accumulate in MFN2KO cells, because of variability effects between the different biological replicates, which prevented the reliable identification of the constituents of the aggregates observed in MFN2KO cells (please see R2 to reviewer #3). This requires many optimization steps and therefore needs to be addressed in future studies.

P3: In Fig. 5 the authors look at several mitochondrial proteins in the 2KO cells by Western-blot analysis, why were these particular proteins selected? What happens with MFN1 in 2KO cells, a Western-blot similar to those in Fig. 5 would be informative?

R3: In fact, the different proteins had been selected for different reasons. To clarify this, we rearranged the manuscript structure and now split the WB analysis to their most logical section. DRP1 and OPA1 analysis aimed at investigating the cross-talk regulation between the other main mitochondrial fission and fusion regulators, given previous reports on their interdependence. OXPHOS and TOM components were analyzed to further confirm their identification in MS analysis as the main pathways affected by MFN2 loss. Finally, PINK1 was used as readout for import defects and due to its impact in mitophagy.

As requested, we have now included a western blot analysis of MFN1 levels in MFN2KO cells (Supplementary Fig. 11d). Despite preventing turnover of MFN1, we did not observe an increase in the steady state levels of MFN1 in MFN2KO cells. This is possibly the net result of both the role of MFN2 in promoting MFN1 turnover and the general decrease in mitochondrial mass observed in MFN2 KO cells.

P4: The authors looked at degradation rate of MFN1 in 2KO and WT cells upon CHX treatment, however did not show if proteasomal inhibition changes the levels of MFN1, as it does for MFN2?

R4: Indeed, as suspected by reviewer #2, proteasomal inhibition changes the levels of MFN1 as it does for MFN2 (new data Supplementary Fig. 11b,c).

P5: OPA1 plays a role both in mitochondrial fusion and fission, dependent on the ratio of the short and long forms of the protein. Was there a shift in the ratio of the short/long forms of OPA1 in 2KO cells?

R5: The levels of the short and long forms of OPA1 in WT, MFN2KO and MFN2KO cells re-expressing MFN2 (named as 2+2 cells) were now separately quantified (Supplementary Fig. 1d). We observe that both the long and short forms of OPA1 are reduced in the absence of MFN2, pointing to a general reduction of the OPA1 protein levels and not to differential processing events.

P6: The authors claim that PINK1 accumulated at the OMM, however show, that epoxomicin inhibition of the proteasome leads to the cleavage of PINK1 by PARL (although to a lower level than in WT cells). What are the levels of PARL in mitochondria in 2KO cells, is the effect on PINK1 cleavage due to the low levels of PARL?

R6: The levels of PARL in MFN2KO cells are now shown in Supplementary Fig. 7d. Importantly, PARL protein levels do not change upon deletion of MFN2. Hence, the increase of full length PINK1 observed in MFN2KO cells is not due to low levels of PARL. This suggestion by the reviewer helped to substantiate the defective import of PINK1 in MFN2KO cells.

P7: Also, the WB analysis in Fig.5 was done on whole-cell extracts, normalized to Ponceau staining. However, if the size/mass of mitochondria was changed in 2KO, this analysis would be invalid. A WB of mitochondrial extracts should be presented.

R7: The referee is indeed correct - mass spectrometry analysis of MFN2KO cells show a decrease of mitochondrial mass in these cells (Fig. 1c). However, after normalization of the mitochondrial mass between all samples (Supplementary Fig. 4c), we still observe reduced levels of the mitochondrial proteins OXPHOS and TOM subunits in MFN2KO cells (Fig. 1d).

To confirm these findings, as requested, we assessed the levels of different mitochondrial proteins in mitochondrial extracts of WT, MFN2KO and 2+2 cells. Consistent with the MS results, we detected increased levels of PINK1 (Supplementary Fig. 7c) and decreased levels of TOM20 (Supplementary Fig. 7b) and OXPHOS (Supplementary Fig. 5c) in MFN2KO cells, and these were restored in 2+2 cells.

P8: It is obvious that the effect of MFN2 ablation on mitochondrial biology is very complex. As mitochondrial import machinery is affected in 2KO cells (Fig. 5), the decrease in other mitochondrial proteins might be due to the decrease in the import, followed by proteasomal degradation of unimported proteins and not really linked to MFN2 directly. Lack of MFN2, as the link to proteasome, should rather lead to accumulation of selective proteins (as the authors show for some proteins) and not to an increased degradation of proteins? Could the authors comment/discuss?

R8: Indeed, MFN2KO represents an extremely complex model of both mitochondrial dysfunction and proteostasis. Our study confirms that MFN2KO cells present mitochondrial dysfunction (fragmented morphology, affected cristae, decreased oxidative phosphorylation), decreased import, and reduced mitochondrial mass. We totally agree that all these aspects can affect each other. Importantly, the many additional experiments performed in this revision to address this very critical question, further support a direct and novel role of MFN2 in constitutive quality

control. On one hand, MFN2 interacts with both HSC70 and the proteasome and cooperates with these machineries to prevent aggregation of newly synthesized proteins (Fig. 5f,g and Fig. 6b). On the other hand, mitochondrial dysfunction upon import blockage does not trigger protein aggregation or MFN2 interaction with the proteasome / HSC70. Moreover, aggregation propensity does not correlate with alterations in mitochondrial morphology (Fig. 7a-c). This is consistent with a direct effect of MFN2 in proteostasis surveillance and not a consequence of its importance in mitochondrial functionality.

P9: The authors show a co-immunoprecipitation of MFN2 with the proteasome. Is the interaction of MFN2 with proteasome lost upon treatment with USP21?

R9: We thank the reviewer for the excellent suggestion of analyzing the role of MFN2 ubiquitylation in its interaction with the proteasome. We failed in deubiquitylating MFN2 in DTT amounts still tolerated by co-immunoprecipitation. Thus, alternatively to using USP21, we opted to address this question on an endogenous level. We depleted protein ubiquitylation in cells by inhibiting the E1 UBA1 with MLN-7243 and performed proximity ligation assays between MFN2 and the proteasomal subunit PSMD11 (Fig. 3e). We indeed found that the interaction of MFN2 with the proteasome is lost in the presence of MLN-7243, showing that MFN2 ubiquitylation is necessary for MFN2 binding to the proteasome (Fig. 3f).

P10: Is MFN2 forming a “link” between mitochondria and proteasome or is the interaction due to the fact that MFN2 is degraded by the proteasome? A very recent paper by this group shows that MFN2 (and especially the CMT2A mutants) is ubiquitinated by UBE4B, and targeted to proteasome, did the authors look at the levels E1/E2/E3/E4 ubiquitin ligases (including UBE4B) in 2KO cells? Are proteasomal subunits and chaperones affected?

R10: Our data indeed indicates that MFN2 forms a link between mitochondria and the proteasome. Importantly, as stated in R9, this interaction depends on MFN2 ubiquitylation. In contrast, MFN1, which is subject to proteasomal degradation with a similar kinetics as MFN2 (Supplementary Fig. 11a,b,c), does not interact with the proteasome (Supplementary 9c). This further supports the specificity of MFN2 and not MFN1 in binding to the proteasome and indicates that being a substrate of the proteasome is not sufficient for a protein to bind to it.

We now show in Supplementary Fig. 12b that the protein levels of UBE4B are largely unchanged in MFN2KO cells. Similarly, the levels of the 20S proteasome, the 19S proteasome subunits PSMC4 and PSMD14, as well as the chaperones/heat shock proteins HSP90, HSC70 and BAG2 are unchanged in 2KO cells (Supplementary Fig. 10b, 13b).

P11: During apoptosis, the authors claim, that MFN1 and PINK1 degradation are prevented in 2KO, while MCL1 is not affected. However, no quantification is provided for PINK1 and MCL1? If MFN2 provides a platform for a specific degradation of proteins by proteasome it would be interesting to analyze the changes in the whole cell proteome under ActD treatment of 2KO vs. WT cells, although I understand this might be beyond the scope of this revision. However, the authors could look at several proteins of OMM (FIS1, MIEF1, DRP1) known to be linked to apoptosis and proteasome.

R11: It would indeed be very interesting to study whole proteome alterations during apoptosis induction. However, as pointed out by the reviewer, this is a costly experiment which we agree falls outside the scope of this revision.

As requested, we have now added quantifications of the protein levels of PINK1 and MCL1 upon apoptosis induction (Fig. 4c,d). Moreover, we have analyzed the levels of several

OMM proteins during apoptosis induction, namely of DRP1, FIS1, MFF, MID49 and MID51 (Response Fig. 5). However, at this point we cannot provide a definite answer to the question from the reviewer because these proteins failed to behave in a reproducible manner, under apoptosis induction with Actinomycin D, in both WT and MFN2KO cells. The reasons for this variability are unclear to us at the moment, especially because MFN1, PINK1 and MCL1 presented a consistent response to Actinomycin D. We therefore feel that including this data in our manuscript would not be advisable.

Response Fig. 5: Western blot quantification of DRP1, FIS1, MFF, MID49 and MID51 protein levels from total cell lysates of HEK WT and 2KO cells untreated (0) or treated with actinomycin D (1 μ M) for 2 or 6h. Bars represent the average fold change relative to untreated WT \pm SD (n=5). Individual values of each experiment are discriminated in white or black filled circles, triangles, squares or diamonds.

P12: Is it possible to use the word “dominance” when describing the fact that MFN2 expression rescues 1KO and DKO? This sentence in the discussion: “These observations underline a dominant role of MFN2 over MFN1 in determining mitochondrial structure in human cells, being MFN1 probably more prominent than MFN2 in hyperfusion events (51–55).”, is not very clear.

R12: We appreciate the reviewer’s comment, and we have now clarified this sentence. It now reads: “These observations underline a broader importance of MFN2 in determining mitochondrial morphology and ultrastructure in human cells. In turn, MFN1 has been more prominently reported to act in stress-induced hyperfusion events.”

P13: The effect of siDRP1 on the resolution of protein aggregates is very intriguing, however this argues against the role of MFN2 in the aggregation? Authors should try to promote mitochondrial elongation/total fragmentation in different ways (CHX treatment, starvation, CCCP) to see if MFN2 or mitochondrial morphology play a role in aggregation.

R13: We took the reviewer’s very nice suggestion and analyzed the effect of disturbing mitochondrial morphology in protein aggregation by multiple cellular treatments. As shown in Fig. 5 e,f,g and Fig. 7a,b,c neither mitochondrial hypertubulation nor fragmentation affected the

outcome of protein aggregation, clearly suggesting that mitochondrial morphology and protein aggregate formation do not correlate.

Since the previous analysis with siDRP1 was based on the very limited number of MFN2KO cells with one mitochondrial tubule, moreover with a very reduced size, we replaced the previous figure by this new data.

P14: MFN2 has been shown to play a role in mitochondria-ER contacts, but there is almost no mention or analysis done in this respect.

R14: To tackle this point, we took advantage of the previously characterized mito-ER tether AKAP1-mRFP-UBC6^{12,13}. Expression of this artificial tether led to a mild decrease of full-length PINK1. However, this was observed in both WT and MFN2KO cells. In addition, the tether did not restore the levels of TOM20 or OXPHOS subunits (Supplementary Fig. 7h). Therefore, although MFN2 has been indeed shown to function as a mitochondrial-ER tether, this property does not seem to be responsible for the accumulation of PINK1 in MFN2KO cells. Unfortunately, we could not analyze the effects of the mito-ER tether in protein aggregation due to incompatibility with the very large excitation/emission wavelength spectrum of Proteostat, similarly to the limitation described in R4 to reviewer 1.

Minor comments:

We are thankful for these minor comments and have corrected them or included them in the manuscript.

MP1: Authors should not use red/green combination for images.

MR1: As suggested, we now present Fig. 1a in the combination of magenta and cyan. In the co-staining figures showing the nucleus, which is labeled with DAPI, in blue, we kept the images in red/green as they are the best combination to have when the color blue is also present.

MP2: Fig. 2b – labels in the figure state WT, MFN1, MFN2 but should be WT, 1KO, 2KO.

MR2: We are grateful to the reviewer for noticing this mistake and have corrected it in the new PCA graph (Fig. 1b).

MP3: Fig. S6A – The size difference between tagged and untagged MFN1/MFN2 seem to be much bigger than in Fig. S5A for MFN2. Did authors use a single FLAG tag, or a multiple FLAG? As MFN1 and tagged-MFN2 run at the same molecular weight in the figure, a blot with anti-FLAG antibody would be useful to confirm that indeed tagged-MFN2 is expressed.

MR3: The old Fig. S6A is now Supplementary Fig. 9a. The apparent difference in the molecular size of FLAG tagged-MFN2 in different Western blots, and consequently better or poorer separation, is due to different percentages of acrylamide in the SDS-PAGE gels. The gel used in old Fig. S6A (now Supplementary Fig. 9a) allows for a much better separation of proteins that run closely together, as is the case of FLAG-tagged MFN2 and endogenous MFN1, as pointed out by the reviewer. We unfortunately were unable to obtain a FLAG decoration of the mentioned blots.

The tag used is a 3xFLAG and the same construct was used throughout the whole paper, in transient transfections and in stable expression.

MP4: Fig. 7a – a legend for the graph is not very clear, as it is not obvious which is WT, 1KO and 2KO, although presumably it lines up with the blot.

MR4: The old Fig. 7a is now Supplementary Fig. 8b. The legend indeed does line up with the blot but to make it more comprehensive we now also labeled the cell lines, as suggested by the reviewer.

MP5: Line 131/132 Taken together, our results confirm the need of both mitofusins for mitochondrial tubulation. – “tubulation” is not the correct word, as they do not tubulate mitochondria, but play a role in fusion.

MR5: We agree and we have corrected it to “mitochondrial fusion”.

MP6: Line 225 rescued all OXPHOS defects of 2KO cells (Supplementary Fig. S6). - Should be Fig. S5.

MR6: We are grateful to the reviewer for noticing this mistake and we have corrected it, now making mention to the new figure number.

MP7: Line 258 proximity to the mitochondria or distributed throughout the cytoplasm (Fig. 3a). - Should be Fig. 3c.

MR7: We thank the reviewer for the warning and we have corrected it, now making mention to a new figure number.

MP8: Line 313 4b, c and Supplementary Fig. 9a). - Should be Fig. 9c.

MR8: We are grateful to the reviewer for noticing this mistake and we have corrected it, now making mention to a new figure number.

MP9: Line 325 Interestingly, the mitochondrial inner membrane fusion factor OPA1 and the mitochondrial fission factor DRP1 were decreased and increased in 2KO cells, respectively, supporting the loss of mitochondrial fragmentation upon MFN2 ablation (Fig. 5a). – I presume the authors mean “supporting the mitochondrial fragmentation upon MFN2 ablation”.

MR9: Indeed, we did mean it as stated by the reviewer.

Reviewer #3 (Remarks to the Author):

This paper presents an exciting story which further substantiates the recent discoveries from the field of proteostasis and organellar quality control linked to the mitochondria. It adds another important player in this process, Mfn2, raising a possibility that this protein is a primarily involved in a process different than mitochondrial fusion.

We are grateful to this reviewer for recognizing MFN2 as a novel player in proteostasis and organellar quality control and for these suggestions, which enabled us to improve the manuscript.

The recent years opened a very interesting and vibrant field of the links between cellular proteostasis and mitochondria, but neither introduction nor discussion reflect it although the current, really interesting and well done study is located placed in this area. Thus, I recommend to modify these parts for better appreciation of this work in the field.

We are thankful for this suggestion and have extensively elaborated the links between cellular proteostasis and mitochondria in the introduction and in the discussion of the manuscript. In the discussion we now added the sub-chapter “*MFN2-dependent proteostasis defects highlight a novel quality control pathway*” dedicated to known mitochondria-cytosolic cooperation pathways in proteotoxic stress.

Major comments

The manuscript provides a comprehensive set of data, the experiments and findings are in general very convincing. The suggested experiments should improve this work:

Point 1 (P1): the most important effects characterizing the proteostasis changes in Mfn2KO (aggregates, a decrease in mitochondrial proteins, i.e. import and the PINK angle) should be performed in the complementation system, to avoid some cross-talk.

Response 1 (R1): We agree with the reviewer that performing the experiments in the complementation system is required. We were indeed lacking the MFN2KO + MFN2 complementation control in the mass spectrometry analysis of the whole cellular proteome and have now added this information in Fig. 1 and Supplementary Fig. 4. This shows a complete rescue of the MFN2KO whole cell proteome alterations by re-expression of MFN2. Additionally, we present rescue experiments regarding the accumulation of protein aggregates (Fig. 5c), the decreased levels of the import machinery (Fig. 2a and Supplementary Fig. 7a,b) and alterations in PINK1 levels (Fig. 2c,d and Supplementary Fig. 7c).

P2: the nature of aggregates should be analyzed by bioinformatic means, with attention to perhaps small qualitative differences from the control with the proteasome inhibitors, especially that the mitochondrial import machinery is decreased in Mfn2KO.

R2: As suggested by the reviewer, we analyzed the mass spectrometry data from the protein aggregate fraction that was presented in the original manuscript. To get some insights into the proteins found within the MFN2KO aggregates, we selected proteins with significant differences in the 3-way ANOVA that were at least 2-fold more abundant in the MFN2KO samples than in the WT samples (previous Supplementary Fig. 8c). These 174 proteins were tested for enrichment of GO-terms and other categories relevant to protein aggregation. No enrichments were observed for propensities to form amyloidogenic aggregates (using Waltz) or phase separations (using ParSe v2). Particularly strong enrichments were observed for nucleoplasmic

proteins (N=107, 3.2-fold enrichment, FDR=10⁻³²), RNA-binding proteins (N=67, 5.2-fold enrichment, FDR=10⁻²⁸) and SUMOylated proteins (N=61, 5.6-fold enrichment, FDR=10⁻²⁶). Interestingly, all these enrichments are also found in epoxomicin-treated cells.

Importantly, however, we noticed a strong inter-replicate variability of the WT (aggregate-free) samples, which hampered the bioinformatical interpretation of the proteins found in the MFN2KO aggregates. As a consequence, hardly any protein showed a significant difference in a two-way t-test between WT and 2KO. This high intra-group variability, together with the small number of proteins specific to 2KO-induced aggregates, indicate that our methods for purifying and analyzing these intracellular structures require further optimization. We therefore feel that including this data in our manuscript would not be advisable.

In fact, the data presented originates from MS analysis of 8M Urea insoluble fractions, which constitutes a highly stringent condition. For this reason, it is possible that fewer proteins present in 2KO insoluble fraction have been retrieved. In the attempt to maximize the isolation of protein aggregates we tested different protocols in what regards the washing of the insoluble pellet obtained by differential fractionation, comparing 8M urea washing to a less stringent solubilizing agent -1% Triton - and no washing (Response Fig. 6). Nevertheless, for the latter two conditions, the amount of protein retrieved from the insoluble fraction (as stained by Ponceau) was extremely high. We therefore did not consider any of these two last conditions for MS analysis.

We feel the point raised by the referee is very important and will optimize the aggregate extraction for a future characterization of the proteome of 2KO protein aggregates.

8M-urea washing:

1%Triton washing:

No washing:

Response Fig. 6: Western blot analysis of soluble and 8M Urea-resistant (upper panel), 1%Triton-resistant (middle panel) or not washed (lower panel) insoluble fractions of untreated HEK WT and 2KO cells, WT cells treated with Epoxomcin (Epoxo) or transfected with Htt-polyQ97 (Q-97-GFP), immunoblotted with anti-ubiquitin (P4D1) and anti-PINK1. Staining of total protein with PoS was used as loading control.

P3: in line with the point above, how the decrease of import machinery affects the aggregation and proteasome binding? how the import deficiency contributes to it?

R3: We thank the reviewer for this suggestion. To understand how import capacity contributes to protein aggregation, we inhibited it with CCCP, and observed that import inhibition does not increase protein aggregation (Fig. 5e and Supplementary Fig. 14b). Since import blockage might repress protein translation due to activation of the integrated stress response¹⁴, we simultaneously treated the cells with CCCP and ISRIB, but this still did not lead to the formation of protein aggregates (Supplementary Fig. 14b).

We also downregulated TOM20, because it was the main TOM subunit affected in MFN2KO cells, but this also did not promote protein aggregate formation in WT cells. However, targeting only one TOM20 component only mildly affected PINK1, which might be insufficient to block protein import (please see Response Fig. 7).

Finally, CCCP treatment decreased the interaction between MFN2 and the proteasome, consistent with the fact that CCCP signals MFN2 proteasomal turnover.

Response Fig. 7: First row: Co-staining of protein aggregation with the PROTEOSTAT® Aggresome detection kit (in red) and TOM20 (in green) in HEK WT cells transfected with scramble (scr) or with siRNA against TOM20 (siTOM20). Bottom row: Western blot analysis of total cell lysates from HEK WT and 2KO, transfected with scr (-) or with siRNA against TOM20 (+), immunoblotted with anti-PINK1 and anti-TOM20. Staining of total protein with PoS was used as loading control. Individual values of each experiment are discriminated in white or black filled triangles or circles. Bars represent the average fold change relative to WT ± SD (n=4).

P4: the proteasome is presented as it would be bound to the surface of mitochondria, based on its interaction with Mfn2. This however is not the case, the large body of the literature shows a lack or no enrichment on the mitochondrial surface under wild type conditions. The situation is changing (slightly) upon stress (the Sladowska et al., and recent Kim et al. 2023). Could authors

comment how they explain the conundrum of seeing proteasome subunits with Mfn2-Flag, given it is properly located in the mitochondrial OM?

R4: We agree that the previous model figure could be misleading and we thank the referee for pointing this out. Indeed, the proteasome is not enriched at the mitochondrial surface, with the exception of a few stress conditions, as mentioned by the reviewer. To show that MFN2 binds to the proteasome and to HSC70, without giving the impression that this is a re-localization of the large body of these components to mitochondria, we added many 26S and HSC70/STIP1/HSC90 to the model figure (Fig. 8).

P5: I am not certain that the name "sentinel" exactly reflects the function of this protein- this is a suggestion for authors consideration.

R5: We have rephrased the text and removed the term "sentinel".

References

1. Mourier, A. *et al.* Mitofusin 2 is required to maintain mitochondrial coenzyme Q levels. *Journal of Cell Biology* **208**, 429–442 (2015).
2. Lee, S. *et al.* Mitofusin 2 is necessary for striatal axonal projections of midbrain dopamine neurons. *Hum Mol Genet* **21**, 4827–4835 (2012).
3. Motori, E. *et al.* Neuronal metabolic rewiring promotes resilience to neurodegeneration caused by mitochondrial dysfunction. *Sci. Adv* **6**, 8271–8299 (2020).
4. Pich, S. *et al.* The Charcot-Marie-Tooth type 2A gene product, Mfn2, up-regulates fuel oxidation through expression of OXPHOS system. *Hum Mol Genet* **14**, 1405–1415 (2005).
5. Bach, D. *et al.* Mitofusin-2 determines mitochondrial network architecture and mitochondrial metabolism: A novel regulatory mechanism altered in obesity. *Journal of Biological Chemistry* **278**, 17190–17197 (2003).
6. Jiang, S. *et al.* Inhibition of mammalian mtDNA transcription acts paradoxically to reverse diet-induced hepatosteatosis and obesity. *Nat Metab* **6**, 1024–1035 (2024).
7. Mourier, A., Ruzzenente, B., Brandt, T., Kühlbrandt, W. & Larsson, N. G. Loss of LRPPRC causes ATP synthase deficiency. *Hum Mol Genet* **23**, 2580–2592 (2014).
8. Hu, L. *et al.* Mfn2/Hsc70 Complex Mediates the Formation of Mitochondria-Lipid Droplets Membrane Contact and Regulates Myocardial Lipid Metabolism. *Advanced Science* **11**, (2024).
9. Meul, T. *et al.* Mitochondrial Regulation of the 26S Proteasome. *Cell Rep* **32**, (2020).
10. Pilecka, I., Sadowski, L., Kalaidzidis, Y. & Miaczynska, M. Recruitment of APPL1 to ubiquitin-rich aggresomes in response to proteasomal impairment. *Exp Cell Res* **317**, 1093–1107 (2011).
11. Chen, H., McCaffery, J. M. & Chan, D. C. Mitochondrial Fusion Protects against Neurodegeneration in the Cerebellum. *Cell* **130**, 548–562 (2007).
12. Göbel, J. *et al.* Mitochondria-Endoplasmic Reticulum Contacts in Reactive Astrocytes Promote Vascular Remodeling. *Cell Metab* **31**, 791-808.e8 (2020).
13. Csordás, G. *et al.* Structural and functional features and significance of the physical linkage between ER and mitochondria. *Journal of Cell Biology* **174**, 915–921 (2006).
14. Kim, J. *et al.* ATAD1 prevents clogging of TOM and damage caused by un-imported mitochondrial proteins. *Cell Rep* **43**, (2024).

Reviewer #1 (Remarks to the Author):

The revised version presented by the authors has responded the major questions raised by the Reviewer, and as a result, the manuscript has improved substantially. I only have some minor issues:

We are very thankful to the Reviewer for acknowledging our efforts to improve the manuscript, and would again like to thank the valid issues raised.

Point 1 (P1): In Table S2 the authors include the protein interactome of both mitofusins. They also show enrichment data for the different proteins expressed as a log₂ ratio MFN1 or 2 and DMSO. They also show statistics, which is shown as $-\log t$ test p value. This is an unusual way to show the statistics and it may be easier to the reader to use p value and adjusted p value.

Response 1 (R1): To facilitate the reader's understanding, as suggested, Table S2 now also shows the p-value and the adjusted p-value (Benjamini Hochberg method).

P2: Analysis of the mitophagy activity shown in Figure 2F should include a group in the presence of bafilomycin A1. Otherwise, they should tone down their conclusions on the impact of MFN2 ablation on mitophagy. In addition, this is not the major observation of this study, and should be omitted from the abstract.

R2: We agree with the Reviewer and we have toned down the conclusions regarding the impact of MFN2 ablation on mitophagy (lines 94, 176, 217, 1279). We also omitted it from the abstract.

P3: Line 402: the authors should state why they used MLN-7243. There is no mention on this, which does not facilitate the reading of the manuscript.

R3: We added a reminder statement to explain the purpose of MLN-7243 usage (line 401-402).

Reviewer #2 (Remarks to the Author):

The authors have now significantly improved the manuscript. They show that MFN2 plays a role in cellular proteostasis, via its interaction with cellular chaperones and with the proteasome. I do have some comments, that need to be addressed before this manuscript is suitable for publication:

We are grateful to the Reviewer for acknowledging the improvement of the manuscript and supporting its publication and we are thankful for the several important points raised.

Point 1 (P1): Authors claim, that Complex II is not affected in 2KO (line 167/168), however their proteomics analysis clearly shows a reduction in protein levels of all subunits of this complex (Supplementary Fig. 5a). Can authors comment?

Response 1 (R1): We are very thankful to the reviewer for having spotted this inconsistency. In the attempt to clarify this point, we studied the protein levels of other CII subunits, namely SDHB and SDHC. Their levels are reduced in 2KO cells (Response Figure 1, below), in agreement with the proteomics data of Supplementary Fig. 5a. Unfortunately, we cannot explain why the SDHA antibody is not consistent with the proteomics analysis. Given that this is not a major point of this manuscript, as previously pointed out by the reviewer 2 (P1 from the 1st revision), we would rather remove the alpha-SDHA western-blot signals from the manuscript (Supplementary Fig. 5b,d and 6a,b), and have adjusted the text accordingly.

Response Fig 1: Western blot analysis and quantification of SDHA, SDHB and SDHC protein levels from total cell lysates of HEK WT, 2KO and 2+2 cells. Bars represent the average fold change relative to untreated WT ± SD (n=3 or 4). Individual values of each experiment are discriminated in white filled or black filled squares, triangles and circles.

P2: Fig. 2f and Supplementary Fig. 6 also shows K357N MFN2 variant, however there is no mention of this in the text?

R2: We thank the reviewer for noticing this. We have now included this information in the manuscript (line 432-434).

P3: Lines 237-240 and Supplementary Fig. 10a: Why is there a staining with MFN2 antibody in 2KO cells that is nuclear? There should not be any PLA signal in 2KO cells, as they do not express MFN2, so this statement is misleading. To show the specificity of MFN2 interaction with proteasome, another OMM protein (such as MFN1) +PSMD11 PLA should be used as a negative control. Did authors perform MFN1-PSMD11 PLA in either WT or 2KO cells? Also, PSMD11 staining in 2KO cells is nuclear, while it is cytosolic in WT, any explanation why?

R3: We thank the reviewer for bringing up this important point and we apologize for the lack of clarity. In fact, in Sup. Fig. 10a, in order not to show only the DAPI signal in the 2KO images, we had increased the gain/intensity, and didn't realize that this was misleading. Now we adjusted the signal intensity the same levels as the images in WT cells. This now clearly shows that there is no PLA interaction between MFN2 and PSMD11 in 2KO cells. We have also done so in Fig. 3 b and f and Supplementary Fig. 12f and 13a and c.

In addition, as requested, we now performed proximity-ligation assay between MFN1 and the proteasome. As shown in the new Sup. Fig. 10b, MFN1 does not interact with 20S. This allows to confirm the mass-spectrometry results from Sup. Fig. 9c and reinforces our conclusions.

P4: In some figures, loading controls are used twice (Supplementary Fig. 5D-SDHA; Supplementary Fig. 7h- Ponceau). Authors should mention this in the legend, or remove one copy, as they might be criticized for image duplication.

R4: We thank the reviewer for raising this important point. In Supplementary Fig. 7h the western-blot signals of NDUFA9 and UQCRC1 belong to the same membrane and therefore, indeed, the Ponceau is the same. This information was now added to the respective figure legend for clarification. Regarding the Supplementary Fig. 5d, we have now removed the western-blot signals for SDHA from the manuscript (please see R3).

P5: Some immunofluorescence images are not very crisp. Could authors improve the contrast? Supplementary F3b, Fig. 3b, f, Supplementary 10a.

R5: We appreciate the reviewer's suggestion for improvement of the immunofluorescence images and have corrected it (please refer also to R3). For Supplementary Fig. 3b, we unfortunately no longer have access to the raw microscopic images.

Reviewer #3 (Remarks to the Author):

I appreciate the current version of the manuscript and fully support it.

We truly appreciate the reviewer for endorsing the publication of the manuscript.